

# Sub-shelf melt pattern and ice sheet mass loss governed by meltwater flow below ice shelves

Franka Jesse[1], Erwin Lambert[2], and Roderik S.W. van de Wal[1,3]

[1]Institute for Marine and Atmospheric Research Utrecht, Utrecht University, Princetonplein 5, 3584 CC Utrecht, The Netherlands
[2]Royal Netherlands Meteorological Institute (KNMI), Utrechtseweg 297, 3731 GA, De Bilt, The Netherlands
[3]Department of Physical Geography, Utrecht University, Utrecht, 3584 CB, The Netherlands

**Correspondence:** Franka Jesse (t.f.jesse@uu.nl)

**Abstract.** Ocean-induced sub-shelf melt is one of the main drivers of Antarctic mass loss. Capturing it in ice sheet models is typically done by using parameterisations that compute sub-shelf melt rates based on local thermal forcing. However, these parameterisations do not resolve the 2D horizontal flow of the meltwater layer, either neglecting it entirely or simplifying its representation. In this study, we present a coupled setup between the ice sheet model IMAU-ICE and the sub-shelf melt model

LADDIE. LADDIE resolves 2D horizontal meltwater flow beneath an ice shelf, incorporating both topographic steering and Coriolis deflection of meltwater plumes. We conduct simulations in a framework closely resembling the Marine Ice Sheet Model Intercomparison Project third phase (MISMIP+), which represents an idealised ice sheet-shelf system. Simulations using LADDIE to calculate melt rates reveal key differences compared to simulations using the widely adopted sub-shelf melt parameterisations. These differences primarily emerge as variations in timing, location, and persistence of strong melting,

leading to distinct transient volume loss. In the LADDIE experiments, a stepwise increase in ocean temperatures induces an initial steepening of the ice draft near the grounding line. This strengthens the westward flow, which converges into a western boundary channel, leading to persistent strong melting along the western margin. Consequently, western margin thinning results in reduced buttressing and strong volume loss over the first period of the simulations. Over longer timescales, a weakened meltwater flow circulation due to reduced thermal forcing at the grounding line allows the western margin to thicken again,

suppressing volume loss. In contrast, the parameterisation's limitations in representing the 2D horizontal meltwater flow prevent these experiments from capturing the influence of ice draft steepening on enhanced margin melt. This results in a different transient volume loss compared to LADDIE. The parameterisations either inherently overestimate the persistence of margin thinning, leading to a sustained strong volume loss, or they underestimate margin thinning, delaying the onset of strong volume loss. Our findings suggest that incorporating the more detailed melt patterns resulting from the 2D horizontal meltwater flow

in ice sheet models could significantly alter projections of Antarctic ice sheet evolution compared to melt patterns computed by the currently more common parameterisations.



# 1 Introduction

The Antarctic ice sheet has been losing mass at an accelerating pace over the last three decades, increasing its contribution to sea level rise (Fox-Kemper et al., 2021; Otosaka et al., 2023). This mass loss is modulated by surrounding ice shelves that buttress the ice flow (Gudmundsson, 2013). Satellite observations indicate a rapid thinning of these ice shelves (Adusumilli et al., 2020; Davison et al., 2023), driven by the enhanced intrusion of warm ocean water into ice shelf cavities causing increased sub-shelf melt (Jenkins et al., 2018). The reduction in ice shelf thickness weakens the buttressing effect, resulting in increased ice discharge from outlet glaciers at the ice sheet margins (Paolo et al., 2015; Rignot et al., 2019). This makes ocean-induced sub-shelf melt one of the main drivers of current Antarctic mass loss (Pritchard et al., 2012; Gudmundsson et al., 2019).

Sub-shelf melting occurs in highly heterogeneous patterns, typically exhibiting higher melt rates near the grounding line and in basal channels due to topographic steering of meltwater plumes (Rignot and Jacobs, 2002; Alley et al., 2016; Berger et al., 2017). Warm ocean water, which is dense and saline, enters the ice shelf cavity and comes into contact with the ice shelf base. Here, meltwater is formed and transported along the ice shelf base before exiting the cavity at the ice shelf front, thereby forming a meltwater layer below the ice shelf (Jenkins, 1991). The interaction between the meltwater flow and the topographic features of the ice shelf base enhances the spatial variability in melt rates, causing certain areas of the ice shelf to thin more rapidly. This local thinning can lead to breakthroughs and collapse of sections of the ice shelf (Alley et al., 2016; Lhermitte et al., 2020; Wearing et al., 2021). Moreover, the distribution of the melt affects the dynamic ice sheet response, with melt along shear-margins inducing a greater decrease in buttressing and a stronger volume loss compared to when the same melt rates are applied along the central flow line (Jordan et al., 2018; Reese et al., 2018b).

One approach to calculate sub-shelf melt to force ice sheet models, involves using 3D cavity-resolving ocean models. Ideally, this is implemented in a coupled setup to capture feedbacks between ice sheet and ocean, such as freshwater fluxes and cavity geometry altering the ocean circulation (De Rydt and Naughten, 2024). Various research groups have coupled ice sheet models to ocean models (Grosfeld and Sandhäger, 2004; Jordan et al., 2018; Naughten et al., 2021; Pelle et al., 2019; Seroussi et al., 2017) or integrated them into Earth System Models (Smith et al., 2021; Siahaan et al., 2022). However, these simulations are computationally expensive and face challenges such as handling cavity expansion due to retreating grounding lines (Fox-Kemper et al., 2019) and bridging resolution gaps between models (Hoffman et al., 2024). As a result, most studies with ice sheet models presently rely on parameterisations to compute sub-shelf melt rates from ocean temperature and salinity profiles (Seroussi et al., 2024). These parameterisations range from a simple linear or quadratic scaling of thermal forcing (Beckmann and Goosse, 2003; Holland et al., 2008; Favier et al., 2019) to more advanced schemes that approximate an overturning circulation in the ice shelf cavity (Reese et al., 2018a; Lazeroms et al., 2018; Pelle et al., 2019), where the upper half of this overturning circulation represents the meltwater layer. While computationally efficient, these parameterisations often require recalibration to adapt to changing ocean conditions and geometries (Favier et al., 2019; Burgard et al., 2022). They also fall short in capturing features like channelised melt due to either neglecting the horizontal component of the meltwater flow, or relying on oversimplified assumptions about its direction.





To improve the representation of 2D horizontal meltwater flow in sub-shelf melt patterns that serve as forcing for ice sheet models, while avoiding the high computational costs of 3D cavity-resolving ocean models, we introduce a new method: an online coupling between the 2D sub-shelf melt model LADDIE (Lambert et al., 2023) and the ice sheet model IMAU-ICE (Berends et al., 2022). LADDIE uses ambient ocean temperature and salinity to resolve the 2D horizontal flow of the meltwater layer beneath the ice shelf, including Coriolis deflection and topographic steering. By resolving these processes, the model captures enhanced melt rates along the western margins and in basal channels, consistent with observations (Zinck et al., 2023).

In this study, we employ our new coupled framework within an idealised domain to examine the effects of 2D horizontal meltwater flow on an ice sheet-shelf system (Asay-Davis et al., 2016). We analyse the feedbacks between ice shelf geometry and melt patterns, as well as their impact on transient volume loss and grounding line retreat. We compare the LADDIE experiments with experiments using three widely used sub-shelf melt parameterisations: the Quadratic parameterisation, scaling melt rates with local temperature (Favier et al., 2019), the PICO parameterisation, an ocean box model simulating the meltwater layer (Reese et al., 2018a), and the Plume parameterisation, describing 1D buoyant meltwater flow (Lazeroms et al., 2019). This comparison allows us to identify the feedbacks missed when 2D horizontal meltwater flow is not resolved and to assess how this affects transient volume loss. We apply two scenarios, wherein ocean conditions transition from a cold state with no melt to either moderate or warm conditions within an idealised framework. These scenarios, applied to both the LADDIE and the parameterised experiments, are used to quantify the response of the grounded ice.

The manuscript is organised into five sections. Section 2 outlines the model setup and experimental design. Section 3 presents our results, divided into two parts: the first focuses on the LADDIE experiments and the melt-geometry feedbacks observed when resolving 2D horizontal meltwater flow; the second examines the parameterised experiments, highlighting the key differences in terms of this feedback between melt and geometry, and their transient volume loss. Section 4 presents the discussion of our findings, emphasising the implications of these differences arising from resolving 2D horizontal meltwater flow compared to using the parameterisations. Finally, Section 5 summarises the main conclusions and recommendations.

## 2 Methods

In Sect. 2.1, we introduce the ice sheet model IMAU-ICE and the sub-shelf melt model LADDIE. Moreover, we discuss the coupling method between these models, including the coupling frequency and extrapolation methods. The coupled setup can be applied on different model domains. Sect. 2.2 introduces the idealised geometry considered in this study, as well as the initialisation, tuning procedure, and details of the individual experiments.



### 2.1 Model description and settings

#### 2.1.1 Ice sheet model IMAU-ICE

We perform ice sheet simulations using the vertically integrated ice sheet model IMAU-ICE v2.0 (Berends et al., 2022). The setup of the model employed in this study uses the hybrid shallow ice / shallow shelf approximation (SIA/SSA) to compute the ice dynamics over floating and grounded parts of the ice sheet. To improve grounding line dynamics, the model scales the basal stress with the sub-grid grounded fraction, following the approach taken by other ice sheet models, like PISM (Feldmann et al., 2014) and CISM (Leguy et al., 2021).

Concerning the default settings for the experiments conducted in this study, we draw upon two key insights from earlier research using this ice sheet model (Berends et al., 2023), which employed a comparable idealised geometry to ours. First, it was demonstrated that the choice of sliding law has little effect on the results for perturbation experiments in an idealised setup. Therefore, we only conduct experiments with a modified power-law relation introduced by Schoof (2005). Second, they evaluated different methods of applying melt near the grounding line and found that the flotation criterion melt parameterisation (FCMP, e.g., Leguy et al. (2020)), which applies melt only to cells meeting the flotation criterion at the centre, is least sensitive to horizontal resolution. Other methods, such as no melt (NMP) or partial melt (PMP), only converge towards FCMP as resolution increases. Based on these findings, we select the FCMP scheme for our experiments.

The standard version of IMAU-ICE includes several sub-shelf melt parameterisations which derive sub-shelf melt rates from ocean data using distinct approaches, particularly in how they represent meltwater flow. The Quadratic parameterisation uses local ocean properties at the base of the ice shelf to compute melt rates and does not account for meltwater flow (Favier et al., 2019). The PICO parameterisation is an ocean box model designed to represent the upper half of the overturning circulation within the ice shelf cavity, corresponding to the meltwater layer (Reese et al., 2018a). It simplifies the meltwater flow by assuming it is always directed from the box near the grounding line towards the box near the ice shelf front. This assumption typically leads to high melt rates in the box nearest to the grounding line and progressively lower melt rates in boxes closer to the ice shelf front. The Plume parameterisation describes the flow of a 1D buoyant meltwater plume and depends on both the local thermal forcing and the thermal forcing at the plume's origin near the grounding line (Lazeroms et al., 2018). In terms of meltwater flow representation, it identifies potential starting points for the plume along the grounding line, based on conditions such as ice shelf base slope and depth, and then assumes a straight 1D flow path from these points. Overall, none of these parameterisations capture the effects of 2D topographic steering and Coriolis deflection of the meltwater flow. Details on the implementation of these parameterisations in IMAU-ICE are given in Appendix A.

#### 2.1.2 Sub-shelf melt model LADDIE

To improve on these built-in parameterisations in IMAU-ICE, we implemented the coupling between IMAU-ICE and the sub-shelf melt model LADDIE (one-layer Antarctic model for dynamical downscaling of ice–ocean exchanges) (Lambert et al., 2023). LADDIE models the well-mixed meltwater layer beneath the ice shelf. By modelling only this meltwater layer,

LADDIE can provide sub-shelf melt fields at a spatial resolution that ice sheet models require while maintaining relatively low



computational costs. The physics incorporated in LADDIE is inspired by earlier work on 2D melt plume models (Holland and Feltham, 2006). The model solves the vertically integrated Navier-Stokes equations to compute horizontal velocities, thickness, temperature, and salinity of the meltwater layer below the ice shelf. By resolving the 2D flow, LADDIE accounts for Coriolis deflection and topographic steering of the meltwater, resulting in more detailed melt patterns than the parameterisations which
either neglect or approximate this flow.

LADDIE is designed to simulate steady-state sub-shelf melt rates. The model domain spans the horizontal extent of the ice shelf (Fig. 1a). The upper boundary of the domain is constrained by the ice shelf draft, required as input for the model. For the lower boundary, LADDIE needs input for the ambient ocean conditions. This can be provided as vertical temperature and salinity profiles, which are then applied uniformly across the horizontal plane.

The thickness of the meltwater layer is affected by vertical exchange of volume at its horizontal interfaces. At the top of the meltwater layer, the vertical exchange of volume, heat and salt is governed by melt or freezing. This is obtained by solving the three-equation formulation for melt and freezing (Holland and Jenkins, 1999; Jenkins et al., 2010). This formulation implies the conservation of heat and salt, and, additionally constrains the ice-ocean boundary to remain at the freezing point. At the bottom of the meltwater layer, the vertical exchange of volume, heat and salt is governed by entrainment of ambient water
into the meltwater layer. In our model setup, we use the Gaspar parameterisation, expressing entrainment as a function of the turbulent velocity in the meltwater layer (Gaspar, 1988; Gladish et al., 2012). For more detailed information on the equations and numerical aspects of the model, we refer to the model description paper (Lambert et al., 2023).

### 2.1.3    Coupling IMAU-ICE and LADDIE

Figure 1a shows a spatial schematic of the coupled setup between LADDIE and IMAU-ICE. At the ice-ocean interface, the
models exchange ice shelf draft geometry (output IMAU-ICE, input LADDIE) and sub-shelf melt rates (output LADDIE, input IMAU-ICE). In our setup, both models use the same grid, such that no interpolation is required.

The coupled simulation procedure can be broken down into four steps (Fig. 1b). At the start of each coupled simulation, both models need to be initialised. This is done by first initialising IMAU-ICE stand-alone (step 1), and then using the resulting initial geometry, along with prescribed ocean forcing, to initialise LADDIE (step 2). Details of the initialisation settings for the
coupled experiments presented in this study are provided in Sect. 2.2.1. Once initialised, the resulting geometry and associated sub-shelf melt rates are used by IMAU-ICE for the first time step of the simulation (step 3). LADDIE is subsequently run for the updated geometry (step 4). Steps 3 and 4 are repeated iteratively until the simulation ends.

The coupling between the two models is asynchronous, meaning that draft geometry and sub-shelf melt rates are exchanged at a certain frequency: the coupling frequency. To ensure that sub-shelf melt rates are compatible with the geometry, the
frequency should be picked such that changes in ice shelf geometry remain minimal over the coupling interval, specifically limiting grounding line retreat to within a single grid cell length. For the experiments conducted in this study, a frequency of 8 times per year is used, resulting into a coupling interval of 0.125 years.

At the start of each coupling interval, LADDIE restarts using the layer thickness, temperature, salinity, and velocities from its last time step of the previous coupling interval, now taking into account the updated geometry from IMAU-ICE. When the





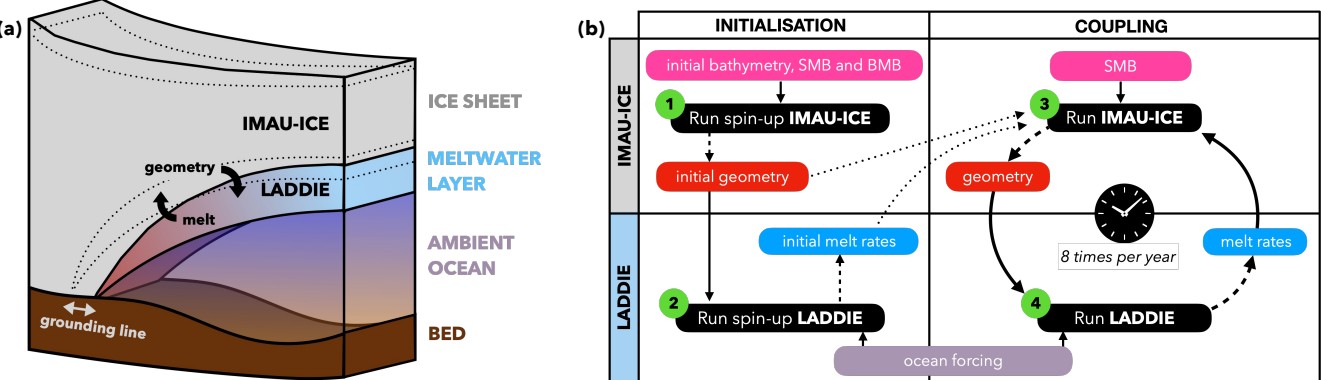

**Figure 1.** Schematic overview of the coupled setup between IMAU-ICE and LADDIE. The spatial overview (a) shows that the models exchange geometry (draft, grounding line and calving front position) and melt rates at the ice-ocean interface. The grounding line is allowed to retreat or advance during the simulation, which affects the horizontal extent of the LADDIE domain. The dotted lines indicate a modified IMAU-ICE geometry, featuring a retreated grounding line, and the associated adjusted lower boundary of LADDIE. In the procedural overview (b), numbers indicate in which order the models run, where step 3 and 4 are repeated until the end of the simulation. The models are coupled asynchronously, meaning that they exchange geometry and melt rates at a fixed coupling frequency (in this case: 8 times per year) which is independent of the time step in the individual models. Dashed arrows indicate model outputs, solid arrows indicate model inputs, and dotted lines indicate model input only used for the first time step of the coupled simulation. SMB and BMB refer to surface mass balance and basal mass balance, respectively.

grounding line retreats, the LADDIE domain expands accordingly, requiring the initialisation of previously grounded cells that are now floating. For the layer thickness, temperature, and salinity, the initial value is set to their nearest neighbour average. The layer velocities are initialised at zero. The required runtime to reach a new quasi-steady-state depends on the flushing time scale, which is influenced by both the size of the ice shelf and the oceanic forcing (Holland, 2017). Therefore, different setups may require varying run durations to reach quasi-steady-state. In our experiments, we run LADDIE for 4 days between each
coupling step to ensure a near-stable meltwater layer thickness and velocity.

As outlined in Sect. 2.1.1, we use the IMAU-ICE setup with the FCMP sub-shelf melt scheme. This scheme applies sub-shelf melt to grounding line cells with a floating centre. However, the model setup is such that LADDIE calculates melt rates only for cells that are fully floating, excluding those partially floating cells that meet the FCMP criterion. To address this discrepancy, we use nearest neighbour averaging to extrapolate the resulting sub-shelf melt field to include the grounding line cells.

## 2.2 Experimental settings

### 2.2.1 Initialisation

The ice sheet model is initialised following the MISMIP+ protocol (Asay-Davis et al., 2016). It represents a symmetric ice stream flowing into a constrained ice shelf (Fig. 2). The model uses a horizontal resolution of 2 km and a dynamic time step,





with a lower limit of 0.01 years and an upper limit of 0.125 years. This variable temporal resolution ensures stability, while at
the same time it limits computation time. The spin-up runs for a total of 50000 years until it reaches a steady-state. It starts
with a 100 m thick slab of ice which grows over time due to the uniform and constant surface mass balance of 0.3 m/year. No
melt at the ice shelf base is applied during the spin-up, ensuring that the initial state is independent of how sub-shelf melting
is prescribed. To obtain a stable central grounding line position at X = 50 km, the spatially homogeneous flow factor is tuned
throughout the spin-up. This flow factor impacts the viscosity field of the ice, consequently influencing the ice velocities. A
fixed calving front is applied at X = 240 km. Ice flowing across the calving front is immediately removed. We do not apply
calving based on a threshold thickness. For all experiments presented in this study, we assume a constant bathymetry.

We initialise LADDIE by prescribing the initial ice sheet geometry along with the scenario-specific ocean temperature and
salinity profiles. The model starts with zero layer velocities, and a uniform layer thickness of 10 meters. The initial temperature
is set to the ambient ocean temperature at the bottom of the meltwater layer, and the initial salinity is set to 0.1 psu below the
ambient ocean salinity, ensuring a stable stratification. We run the model for 40 days to obtain a quasi-steady-state melt pattern
that aligns with the initial ice sheet geometry and forcing data. Specific parameter settings for IMAU-ICE and LADDIE are
provided in Appendix B.

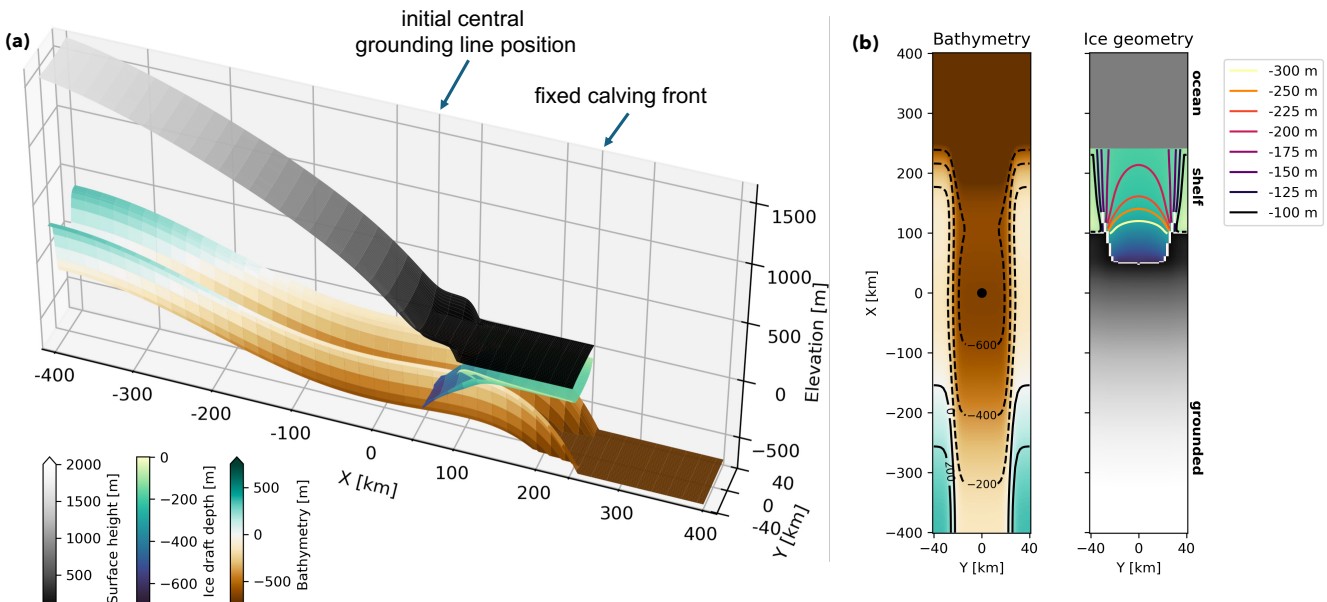

**Figure 2.** The idealised MISMIP+ geometry. (a) 3D view of bathymetry, surface height, and ice draft depth. The initial grounding line is at
X = 50 km, and a fixed calving front is applied at X = 240 km. (b) Top view of the bathymetry and the ice geometry, using the same colour
scales and colour maps as in panel (a). The bathymetry top view includes contours for various depth levels, highlighting the lowest point
from the initial grounding line inwards, indicated by the black dot at X = 0 km. The ice geometry shows the grounding line in white and ice
draft contours at levels specified in the legend.



### 2.2.2 Perturbation experiments

In this study, we aim to improve the understanding of the transient response of an ice sheet-shelf system to sub-shelf melt
patterns that are governed by a 2D horizontal meltwater flow. Additionally, we compare this to the transient response of the
system to more idealised melt patterns computed by sub-shelf melt parameterisations. These objectives guide the selection
of the perturbation experiments presented in Table 1. All simulations start from the same initial geometry (Fig. 2), and are
run over a period of 1000 years. At the start of the simulations, sub-shelf melt is turned on, either through the coupling with
LADDIE, or through turning on one of the built-in sub-shelf melt parameterisations. We consider three such parameterisations:
Quadratic, PICO and Plume, as described in Sect. 2.1.1. In addition to the perturbation experiments, we include a 'no melt'
control run to evaluate model drift. This control run is solely used for drift assessment and is not employed for corrections to
the perturbation experiments.

| Experiment name | Sub-shelf melt implementation | Forcing scenario |
| --- | --- | --- |
| LA_M | LADDIE | moderate |
| QU_M | Quadratic | moderate |
| PI_M | PICO | moderate |
| PL_M | Plume | moderate |
| LA_H | LADDIE | high |
| QU_H | Quadratic | high |
| PI_H | PICO | high |
| PL_H | Plume | high |
| control | - | no melt |

**Table 1.** Overview of experiments. The forcing scenario relates to the prescribed temperature and salinity profiles as shown in Fig. 3.

To test the sensitivity to different ocean temperature states, we apply two forcing scenarios: a moderate-melt scenario and a
high-melt scenario (Fig. 3). Both scenarios are expressed as a tangent hyperbolic profile of temperature, resembling temperature
profiles observed around Dronning Maud Land, and the Amundsen Sea (Locarnini et al., 2018). The salinity profiles are
determined such that the density profiles of the moderate-melt and high-melt scenario are identical. The forcing is constant
over the entire duration of the simulations, allowing us to isolate the feedback between melt patterns and ice shelf geometry
under steady ambient ocean conditions.

### 2.2.3 Tuning

To ensure a valid comparison between the simulations, we tune the heat exchange coefficient ($\gamma_T$) such that for the moderate-
melt scenario, and based on the initial geometry, the average melt rates over the deep area of the ice shelf are consistent between
LADDIE and the parameterisations. Here, we define the deep area as the area where the ice shelf draft is below $-300$ meters
(yellow contour in Fig. 2b), following the definition in the ISOMIP+ protocol (Asay-Davis et al., 2016).



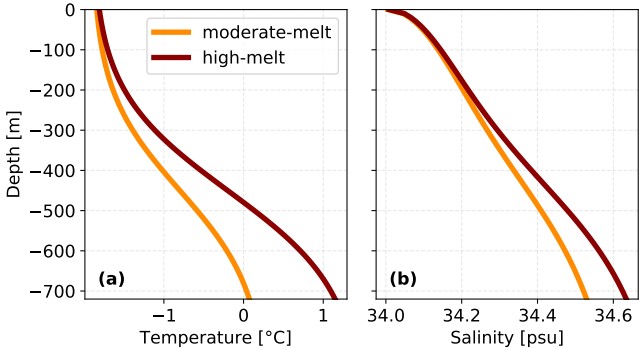

**Figure 3.** Ambient ocean temperature (a) and salinity (b) profiles for the two forcing-scenarios used in the experiments. The vertically varying profiles are uniform in horizontal space and remain constant over time.

For LADDIE, we first tune $\gamma_T$ such that the average deep melt rates in the high-melt scenario are within the ISOMIP+
target of $30 \pm 1$ m/year, determined for their WARM scenario. We then apply the resulting heat exchange coefficient to the
moderate-melt scenario, resulting in average deep melt rates of 16.75 m/year. Earlier work shows that no retuning is required
for LADDIE (Lambert et al., 2023), so we take the same value of $\gamma_T$ for both forcing scenarios.

We then take the resultant averaged deep melt rates in the moderate-melt scenario to tune the parameterisations. For PICO,
following Reese et al. (2023), we additionally tune the overturning coefficient ($C$) to optimally match the integrated melt. The
values obtained from our tuning procedure are presented in Table 2.

| Sub-shelf melt implementation | $\gamma_T$ [ms$^{-1}$] | $C$ [m$^3$ kg$^{-1}$] |
|---|---|---|
| LADDIE | $1.47 \times 10^{-4}$ | – |
| Quadratic | $1.42 \times 10^{-3}$ | – |
| PICO | $1.28 \times 10^{-4}$ | $0.23 \times 10^{-6}$ |
| Plume | $1.69 \times 10^{-3}$ | – |

**Table 2.** Values of the heat exchange coefficient ($\gamma_T$) and overturning coefficient ($C$) used for tuning the different sub-shelf melt implementations.

## 3    Results

In Sect. 3.1, we examine the feedback between sub-shelf melt pattern and ice sheet geometry for the LADDIE experiment
with moderate-melt forcing (LA_M). In Sect. 3.2, we compare the LADDIE experiments with the parameterised experiments,
beginning with the initial melt patterns (Sect. 3.2.1), followed by the melt-geometry feedback (Sect. 3.2.2), and concluding
with the volume above flotation and grounding line retreat rates (Sect. 3.3.3).



## 3.1 Melt-geometry feedback in LADDIE experiments

The simulation with LADDIE melt and moderate-melt forcing (LA_M) shows a rapid grounding line retreat over the first period of the simulation, initiated by high melt rates near the central grounding line and along the western boundary (Fig. 4). The distribution of high melt rates leads to a steepening of the ice shelf base near the central grounding line. This accelerates the plume flowing along the central and western grounding line, deepening the western boundary channel and enhancing deep melt rates. On longer time scales, the ice shelf flattens and the grounding line retreats and resides at shallower depth. As a result, the meltwater flow circulation weakens and melt rates reduce, enabling the ice shelf to stabilise on the lateral margins. In the next paragraphs, we describe this evolution in more detail.

The initial LADDIE melt pattern reveals enhanced melt rates, not only near the deep central grounding line, but also along both the shallower western and eastern flanks (Fig. 4b). The meltwater layer velocities show that, right at the grounding line, the meltwater flow rises along the ice shelf draft following the ice flow direction, but then curves towards the west due to Coriolis deflection (Fig. 4c). This westward flow converges into a western boundary current which flows along the western margin of the ice shelf until it eventually reaches the ice shelf front. The boundary current leads to enhanced melt rates on the western side of the shelf, also close to the western margin (WM) grounding line. Additionally, the meltwater layer velocities show a current originating at the eastern part of the grounding line. Guided by topographic steering, this current follows the draft contours towards the ice shelf front (Fig. 4a, c), leading to enhanced melt rates on the eastern side of the ice shelf. Melt rates near the eastern margin (EM) grounding line remain close to zero, however.

Over the first 200 model years, ice draft steepening near the central grounding line, driven by earlier melt patterns and accompanied by substantial retreat, increases layer velocities at this location (Fig. 4e, g). This velocity increase affects the melt pattern in two ways (Fig. 4f): first, the increase in velocity directly raises local melt rates; second, it enhances Coriolis deflection of the flow, reinforcing the western boundary channel relative to the eastern boundary channel. These combined effects concentrate high melt rates closer to the grounding line and increase deep melt rates compared to the initial state (Fig. 4b, f). Over this 200-year period, the feedback between ice draft steepening and the melt pattern becomes self-sustaining, as the enhanced melt near the grounding line maintains the steep ice shelf draft.

The reinforcement of the western current relative to the eastern current results in an asymmetry in the locations of the WM and EM grounding lines (Fig. 4e). The ice velocity field reveals stronger velocity gradients downstream of the WM grounding line than those observed downstream of the EM grounding line (Fig. 4h). Near the WM grounding line, enhanced thinning has created a zone where the ice becomes too thin to sustain viscous flow. This effectively causes the grounded ice to dynamically detach from the floating ice, meaning that the velocities in the grounded region are isolated from the influence of the downstream floating ice. This dynamical detachment results in steep velocity gradients and near-zero velocities at the WM grounding line, allowing it to maintain its position. In contrast, at the EM grounding line, a smoother transition from grounded ice to floating ice enables the ice to dynamically thin and eventually unground, causing the EM grounding line to retreat over a distance of 70 km over the first 200 years.



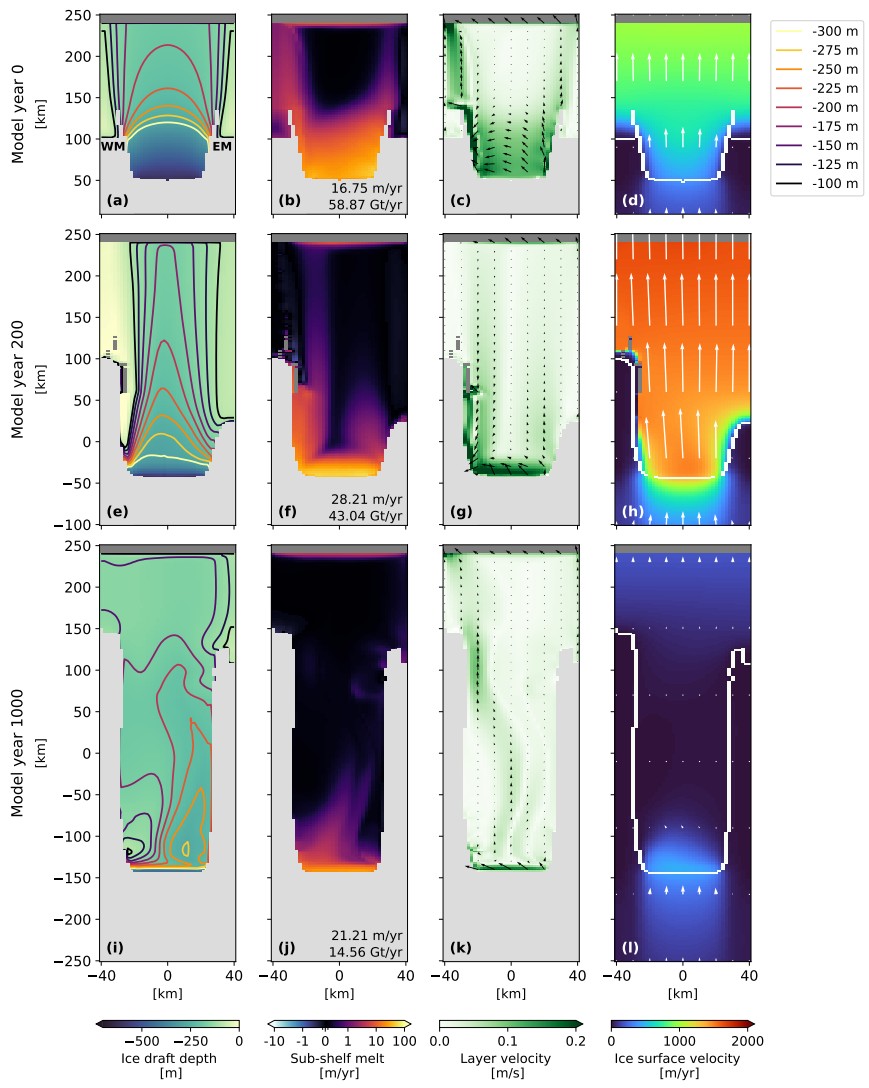

**Figure 4.** Transient response of main variables in the LADDIE experiment with moderate-melt forcing (LA_M). The columns present the ice draft geometry, sub-shelf melt rates, meltwater layer velocities, and ice surface velocities. Contours in the first column represent draft depth levels, as indicated in the legend. The average deep melt rates and integrated melt flux are given in the lower right corner of the sub-shelf melt panels. In the third and fourth columns, arrows illustrate the direction and magnitude of the meltwater layer velocity and ice surface velocity, respectively. In the ice surface velocity plots, the grounding line is marked by a white line. Each row corresponds to a different time slice, indicated to the left of each row. The western margin and eastern margin are labelled as WM and EM in the upper left panel. Grounded ice is masked in light grey in the first three columns. In all panels, the ocean is masked by dark grey. Animations of this figure, showing time slices every 10 model years, are provided in the video supplements.





The persisting western boundary current causes a localised collapse along the western margin, as the ice flow convergence
cannot compensate for the high melt rates in these areas (Fig. 4e). When the western boundary current encounters one of these
gaps in the ice shelf, the buoyant plume exits the cavity through these openings (assumption in LADDIE). The gap in the ice
shelf allows heat to escape the cavity and the discontinuous ice shelf draft prevents the continuation of the boundary current
downstream from the gap. This localised collapse can thus serve as a negative feedback, which limits the impact of warmer
forcing.

Additionally, we note that the integrated melt flux is reduced relative to the initial state (Fig. 4b, f). Both the ice shelf area
and deep melt rates increased, indicating that the integrated melt over the shallower part has reduced substantially. We attribute
this, first, to a larger part of the ice shelf now residing in shallower waters, where ambient ocean temperatures are colder, and
hence melt rates are lower. Second, we attribute it to large parts of the ice shelf becoming more flat, which suppresses layer
velocities, consequently lowering melt rates.

After 1000 years, further retreat of the central grounding line leads to a reduced thermal forcing at this location, as it now
resides in shallower, hence colder, waters (Fig. 4i). This results in two major changes to the melt pattern: first, it reduces
melt rates near the central grounding line; second, it reduces melt along the western margin, as the reduced thermal forcing is
insufficient to maintain the strong western boundary current (Fig. 4j). The weakened circulation, combined with topographic
constraints imposed by the ice shelf draft, forces the channelised melting originating at the central grounding line to migrate
eastwards (Fig. 4k). Melt rates associated with this eastward shifted channel remain low, however (Fig. 4j). Moreover, the
weaker meltwater flow, accompanied by reduced deep and integrated melt rates, allows for ice thickening along both the eastern
and western margins. This, in turn, enables the readvance of the WM and EM grounding lines, which increases buttressing of
the ice flow. Together with the lower melt rates in general, the increased buttressing reduces the ice flow substantially compared
to the model year 200 (Fig. 4h, l).

The response in the high-melt scenario experiment using LADDIE melt (LA_H) is qualitatively similar to the LA_M ex-
periment, although some differences in timing are observed (Appendix C, Fig. C1). The fast retreat period is shorter in the
high-melt scenario due to earlier and more frequent local collapse of the ice shelf causing the western boundary current to
lose momentum and melt potential. Additionally, the eastward shift of the channelised meltwater flow occurs earlier in the
high-melt scenario, allowing the channel to migrate fully to the east by the end of the 1000-year simulation. Similar to the
LA_M experiment, this channel is associated with low melt rates.

## 3.2 Comparison with sub-shelf melt parameterisations

### 3.2.1 Initial melt fields

The initial melt patterns of the parameterised experiments reveal three key differences when compared to the initial melt
patterns from the LADDIE experiments (Fig. 5). These differences are largely consistent between the moderate-melt and high-
melt scenario, and relate to (1) melt rates along the western and eastern flanks of the ice shelf, (2) the distribution of melt near
the central grounding line, and (3) the integrated melt flux.





First, melt rates along the western and eastern flanks of the ice shelf differ across the sub-shelf melt implementations, both in the deep tuning zone and in shallower areas. For LADDIE, the Coriolis deflection and topographic steering of the meltwater flow leads to greater melt along the margins than in the centre (Fig. 5a). In contrast, the Quadratic and Plume parameterisations show a much more centralised melt distribution, with minimal melt along the flanks and EM and WM grounding lines (Fig. 5b, d). For the Quadratic parameterisation, this is related to these areas being shallow and therefore experiencing low thermal forcing. For the Plume parameterisation, the prescribed flow pathway appears to underestimate the flow convergence towards the margins, compared to when the 2D flow is resolved (as in LADDIE). The PICO parameterisation shows enhanced melt rates along the margins (Fig. 5c), similar to LADDIE, but for a different reason. PICO assumes meltwater flow origin along the entire grounding line, whereas in LADDIE, the plume origin is determined by entrainment which is strongest in the deepest regions. This difference causes PICO's melt rates along the margins, beyond the deep tuning area, to exceed those of LADDIE. Additionally, PICO shows enhanced melt at both the WM and EM grounding lines. This contrasts with LADDIE's pattern,

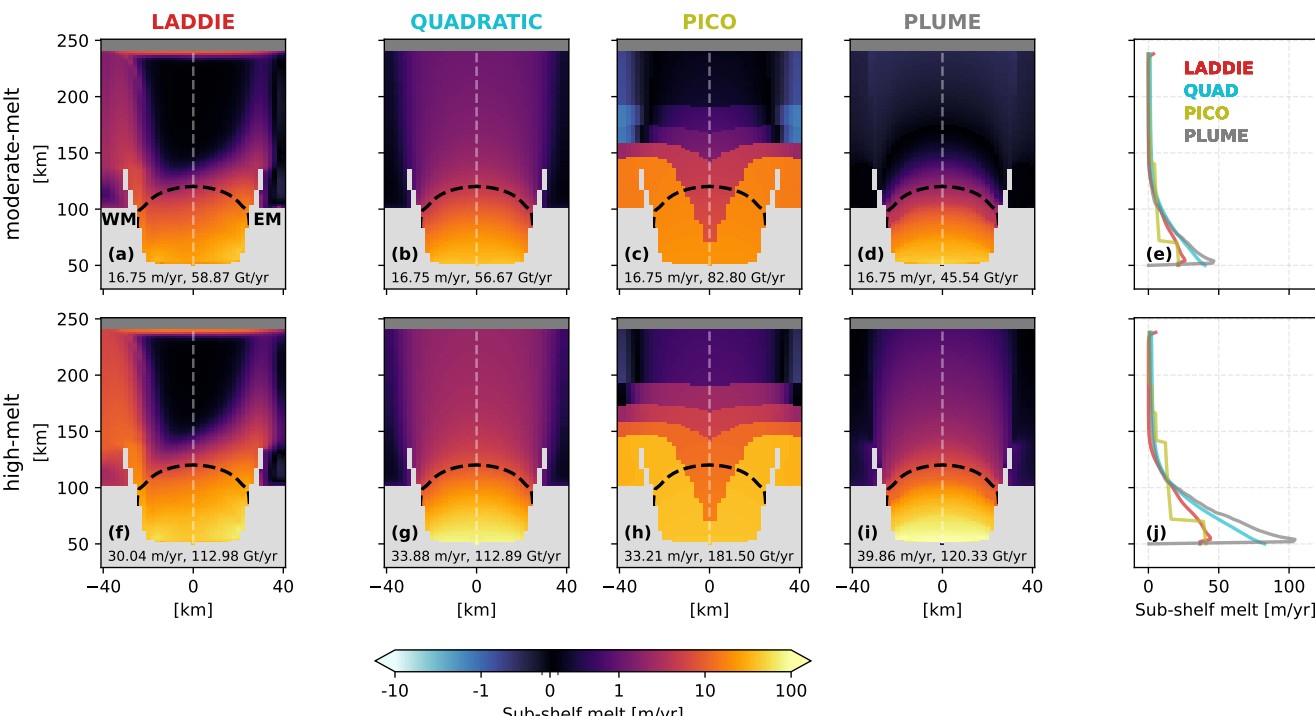

**Figure 5.** Initial melt patterns for the moderate-melt (a-d) and high-melt scenario experiments (f-i) resulting from the tuning procedure (Sect. 2.2.3). The dashed line denotes the 300 m ice shelf draft contour, indicating the deep part of the shelf on which tuning is based. Colours indicate melt rates in m/yr. The grey area indicates grounded ice and the blue zone indicates the ocean (so no ice). The averaged deep melt rates (in m/yr) and integrated melt flux (in Gt/yr) are given at the bottom of each panel. The right-most panels show melt rates along the central flow line (indicated by grey dashed line in the top view plots) for the different sub-shelf melt parameterisations in the moderate-melt (e) and high-melt (j) scenarios.





where enhanced melt is only observed at the WM grounding line, where heat transported by the western boundary current can enter the region.

Second, the melt rates along the central flow line reveal a different spatial distribution within the deep tuning area, in particular close to the grounding line (Fig. 5e, j). The Quadratic and Plume parameterisations show substantially higher melt rates close to the grounding line, compared to LADDIE. In contrast, the PICO parameterisation shows melt rates near the grounding line that are more similar to LADDIE's. However, PICO's box structure results in a much more abrupt decline in melt rates along the central flow line, highlighting its spatially segmented approach, which contrasts with the more gradual

reduction observed in LADDIE, Quadratic, and Plume.

    Third, the variations in integrated melt flux reflect differences in how melt is distributed between the deeper and shallower regions of the ice shelf, given that average melt rates in the deep tuning area are consistent across all four sub-shelf melt implementations in the moderate-melt scenario. The Quadratic parameterisation shows an integrated melt flux similar in magnitude to LADDIE for both scenarios (Fig. 5b, g), implying a comparable distribution of melt between deep and shallow areas. The

PICO parameterisation exhibits a larger integrated flux compared to LADDIE (Fig. 5c). This is due to the high melt rates assigned to the shallow grounding lines along both margins in PICO's box structure, resulting in more melt occurring in shallower regions. The Plume parameterisation shows the smallest integrated flux (Fig. 5d), with only little melt outside the deep tuning area, indicating lower melt rates in shallow areas relative to LADDIE. Note that, in the high-melt scenario the Plume parameterisation produces a larger integrated flux than LADDIE (Fig. 5f, i). This is mainly explained by the increased deep

melt rates relative to LADDIE.

    Lastly, we remark that the parameterisations exhibit a greater sensitivity to warming, compared to LADDIE. This follows from the deep melt rates for the high-melt scenario for the parameterised melt fields (Fig. 5g, h, i), exceeding the LADDIE deep melt rates (Fig. 5f). The Plume parameterisation has the highest sensitivity, followed by the Quadratic and PICO parameterisations, which have a similar sensitivity.

**3.2.2  Melt-geometry feedback**

In this section, we discuss the feedback between the ice shelf geometry and the sub-shelf melt pattern for the parameterised moderate-melt experiments, and compare it to the LA_M experiment (Sect. 3.1). Figure 6 shows the ice shelf draft, sub-shelf melt pattern and ice surface velocities after 200 and 1000 model years for the QU_M, PI_M, and PL_M experiments. These experiments agree with the LA_M experiment on an initial speed up of the ice flow, and a final stabilisation. The magnitude

of ice speed-up and the stabilising mechanisms differ from the LA_M experiment. These differences are related to variations in timing, location, and persistence of enhanced melt, where persistence is defined as the duration over which high melt rates are maintained in a specific region before decreasing again. The following paragraphs provide a more detailed discussion of the feedback between the melt pattern and the geometry for each parameterisation, and how it compares with LADDIE. The high-melt scenario results are qualitatively similar (Appendix C), hence we only discuss the moderate-melt experiments here.

In agreement with the LA_M experiment, the QU_M experiment shows steepening of the ice draft near the central grounding line over the first 200 model years, indicated by the draft contours moving closer to the central grounding line (Fig. 6a). Unlike



for LA_M, this steepening is accompanied with suppressed deep melt rates (Fig. 6b). These suppressed deep melt rates are caused by the onset of melt, which induces ice shelf thinning, resulting in a shallower ice draft which is in contact with colder waters. Additionally, in QU_M, both the EM and WM grounding lines retreat over the first 200 model years (Fig. 6a). This differs from LA_M, where only the EM grounding line retreats due to dynamical detachment at the WM (Fig. 4e). In QU_M, the low melt rates are insufficient to induce a dynamical detachment. Rather, the relatively thick ice maintains a smooth gradient in ice velocity along both margins (Fig. 6c), allowing both margin grounding lines to retreat.

Towards the end of the QU_M experiment, the ice shelf pins on the shallow margins, increasing buttressing and reducing ice velocities (Fig. 6j, l). The cause of pinning differs from the cause of readvance in LA_M. For QU_M, pinning at the margins

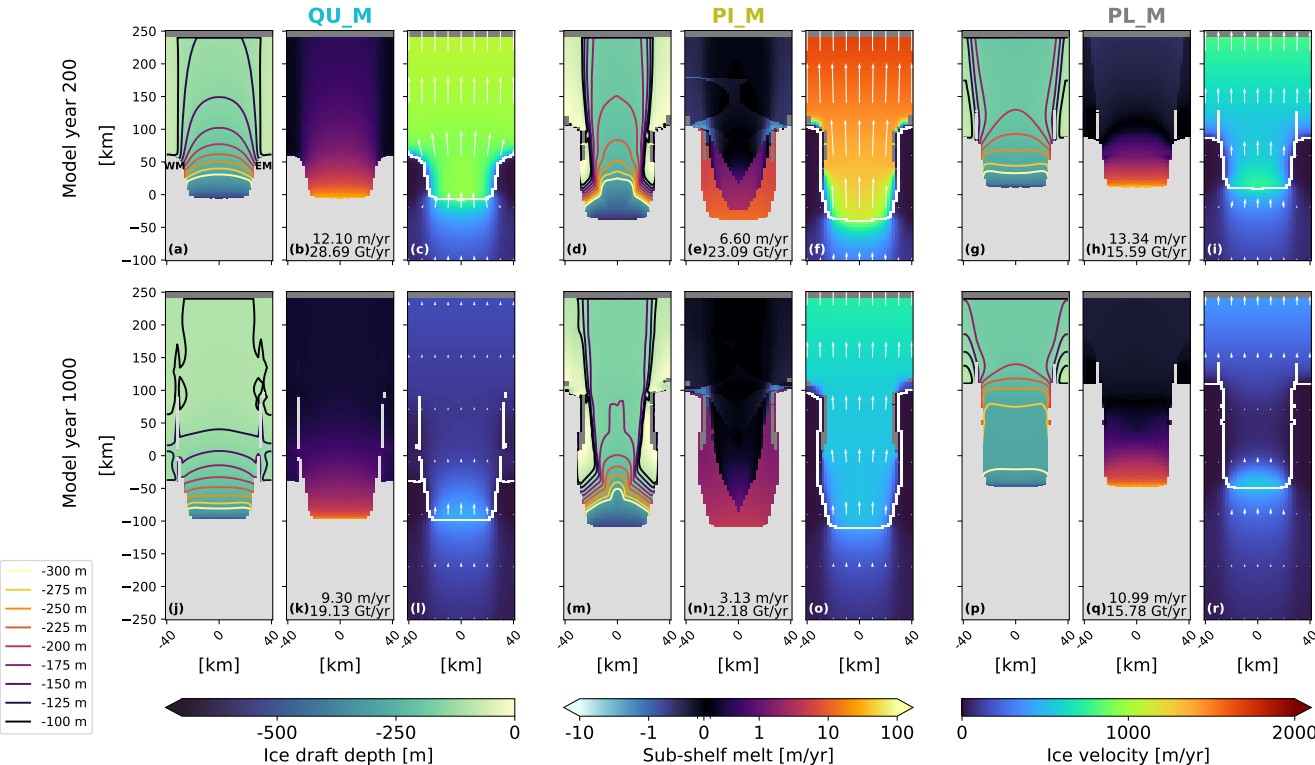

**Figure 6.** Transient response of main variables in the parameterised experiments with moderate forcing (QU_M, PI_M, PL_M). The columns present the ice draft geometry, sub-shelf melt rates, and ice surface velocities. Each row corresponds to a different time slice, indicated to the left of each row. Contours in the first column represent draft depth levels, as indicated in the legend. The average deep melt rates and integrated melt flux are given in the lower right corner of the sub-shelf melt panels. In the ice surface velocity plots, arrows illustrate the direction and magnitude of the ice surface velocities, and the white line denotes the grounding line. The western margin and eastern margin are labelled as WM and EM in the upper left panel. Grounded ice is masked in light grey for the ice draft geometry and sub-shelf melt rates. In all panels, the ocean is masked by dark grey. Animations of this figure, showing time slices every 10 model years, are provided in the video supplements.





occurs due to thickening, which results from low thermal forcing in these specific areas, caused by the shallow ice draft. In contrast, for LA_M, the thickening at the margins is driven by reduced thermal forcing at the central grounding line, which leads to the migration of the western boundary channel and lower melt rates along the eastern margin. In summary, for QU_M, pinning on the margins is driven by local processes, as this parameterisation neglects meltwater flow, while for LA_M, the readvance is driven by the larger-scale meltwater dynamics.

The PI_M experiment shows high melt rates persisting on both sides after 200 model years (Fig. 6e), in contrast to the LA_M experiment, where melt persists along the western margin but weakens along the eastern margin. The sustained melt on both sides in the PI_M experiment keeps both margins thin and regionally causes ice shelf collapse (Fig. 6d). The thin margins allow for dynamical detachment on both sides of the ice shelf (Fig. 6f), similar to that observed at the WM in the LA_M experiment (Fig. 4h). The dynamical detachment in PI_M has two consequences: first, the EM and WM grounding lines remain at their original position as the ice on the margins is no longer stretched by the downstream ice shelf. Second, it reduces buttressing, allowing for a more rapid ice flow despite the extensively grounded margins.

After 1000 model years, the margins remain thin in the PI_M experiment due to persistent margin melt (Fig. 6m, n). This contrasts with LA_M, where margin thinning does not persist over this time scale (Fig. 4i, j). Despite the persistent margin thinning in PI_M, ice velocities decrease compared to the model year 200 (Fig. 6f, o). These reduced velocities are partly due to decreased thermal forcing at the shallower grounding line, which lowers overall melt rates. Additionally, they are caused by dynamical reattachment near the WM and EM grounding lines, which slightly increases buttressing compared to earlier. However, ice velocities in PI_M remain higher than those observed at the end of the LA_M experiment (Fig. 4l), as the weaker margins still allow for a faster ice flow.

The PL_M experiment shows ice draft steepening near the grounding line accompanied by a reduction in average deep melt rates over the first 200 model years (Fig. 6g, h), similar to QU_M but contrasting with LA_M. Like LADDIE, the Plume parameterisation relies on ice draft steepness to compute plume velocities. In LADDIE, ice draft steepening establishes a meltwater current along the draft contour line, which enhances melt along this line and further steepens the gradient. This creates a positive feedback between deep melt rates and grounding line steepening. In contrast, the Plume parameterisation generally directs meltwater flow across the draft contour lines (away from the grounding line), preventing the parameterisation to represent this positive feedback. Additionally, while the margin grounding lines have retreated slightly, part of the ice shelf remains pinned on the shallow margins, resulting in lower ice velocities compared to the other experiments (Fig. 6i).

At the end of the PL_M simulation, the EM and WM grounding lines readvance, which suppresses the ice velocities (Fig. 6p, r). The readvance is driven by low thermal forcing at the origin of the plume, located at the shallow EM and WM grounding lines. This low thermal forcing results in limited melt below both flanks of the ice shelf, allowing for regrowth and regrounding (Fig. 6q). The limited melt below the flanks was also observed in the later stages of the LA_M experiment, where it similarly causes readvance of the margins. However, in the Plume parameterisation, the shallow lateral margin grounding lines are considered the origin of the plume. As a result, the readvance occurs earlier in the simulation compared to LADDIE, which uses the deeper grounding line as the plume origin.





To synthesise, the Quadratic and Plume parameterisations respond similarly to melt onset. Both capture an initial ice shelf draft steepening, but they fail to represent the positive feedback between enhanced deep melt rates and draft steepening governed by the horizontal meltwater flow adjustment. Over longer time scales, low melt rates along the margins enable stabilisation on these margins through pinning (Quadratic) or a combination of pinning and readvance (Plume). Although this stabilisation resembles that simulated by LADDIE, it arises from different mechanisms tied to the parameterisation's limitations in representing the 2D meltwater flow. For the PICO parameterisation, the melt onset causes thinning and a partial collapse of the lateral margins, reducing buttressing in the short term. This behaviour resembles LADDIE's western margin response but arises from PICO's assumption of meltwater flow origin along the entire grounding line, as opposed to LADDIE's plume originating near the deep grounding line and flowing along the grounding line. Over time, this persistent margin thinning prevents stabilisation, leading to higher ice velocities in PICO compared to LADDIE, where margins thicken due to a weaker meltwater flow.

### 3.2.3 Volume above flotation and grounding line retreat

The previous sections covered a qualitative discussion of the melt-geometry feedback for the different moderate-melt simulations. Here, we explore the implications of this feedback on the volume above flotation (VAF) and changes in central grounding line position. By considering these variables, we can quantitatively compare the transient responses of the different simulations and gain deeper insights into their sensitivity to different forcing scenarios. The control experiment, with no melt applied, shows VAF loss rates close to zero throughout the entire simulation, indicating a negligible model drift (Fig. 7a, b).

The choice of sub-shelf melt implementation affects the timing and magnitude of peak VAF loss in both moderate-melt and high-melt scenarios (Fig. 7a, b). In the LADDIE experiments, VAF loss gradually intensifies over the first 100 years, driven by the positive feedback between ice draft steepness and deep melt rates and by thinning along the western boundary. The difference in the duration of strong VAF loss between the moderate-melt and high-melt scenario is explained by the reduced persistence of the western boundary channel in the high-melt scenario due to more frequent ice shelf collapse (Appendix C). The Quadratic experiments show a delayed peak VAF loss, relative to the LADDIE experiments. The peak is driven by the retreat of the EM and WM grounding lines, which reduces buttressing and subsequently increases ice velocities later in the simulation. This process of reducing buttressing is slower than the dynamical detachment in the LADDIE experiments, explaining the delay in the peak VAF loss. In contrast, the PICO experiments show an earlier VAF loss peak compared to the LADDIE experiments. This is due to a more rapid margin thinning, inducing earlier dynamical detachment leading to reduced buttressing. The loss rates reduce after about 100 years, when the grounding line retreats past the overdeepening in the bedrock. Past this overdeepening, reduced thermal forcing lowers melt rates, resulting in a gradual decrease in VAF loss rates. Lastly, the Plume experiments show relatively constant VAF loss rates without pronounced peaks. This steady loss is due to persistent pinning along the margins, which restricts ice flow acceleration in the moderate-melt scenario and only moderately allows for it in the high-melt scenario.

Over the first 200 model years, the distinct transient behaviour divides the experiments in two groups, with the LADDIE and PICO experiments showing a stronger VAF loss than the Quadratic and Plume experiments (Fig. 7c, d). This division is also





reflected in the grounding line retreat rates over the first 100 model years (Fig. 7e, f). While the initial melt patterns show the highest melt rates near the grounding line for the Plume and Quadratic parameterisation (Fig. 5e, j), the grounding line retreat

rates show that these near-grounding line melt rates are not a reliable indicator for the retreat rate. Instead, margin weakening and dynamical detachment allow for a stronger ice speed up in LADDIE and PICO, hence a stronger grounding line retreat. For Quadratic and Plume, strong buttressing prevents a rapid grounding line retreat, regardless of the melt near the grounding line.

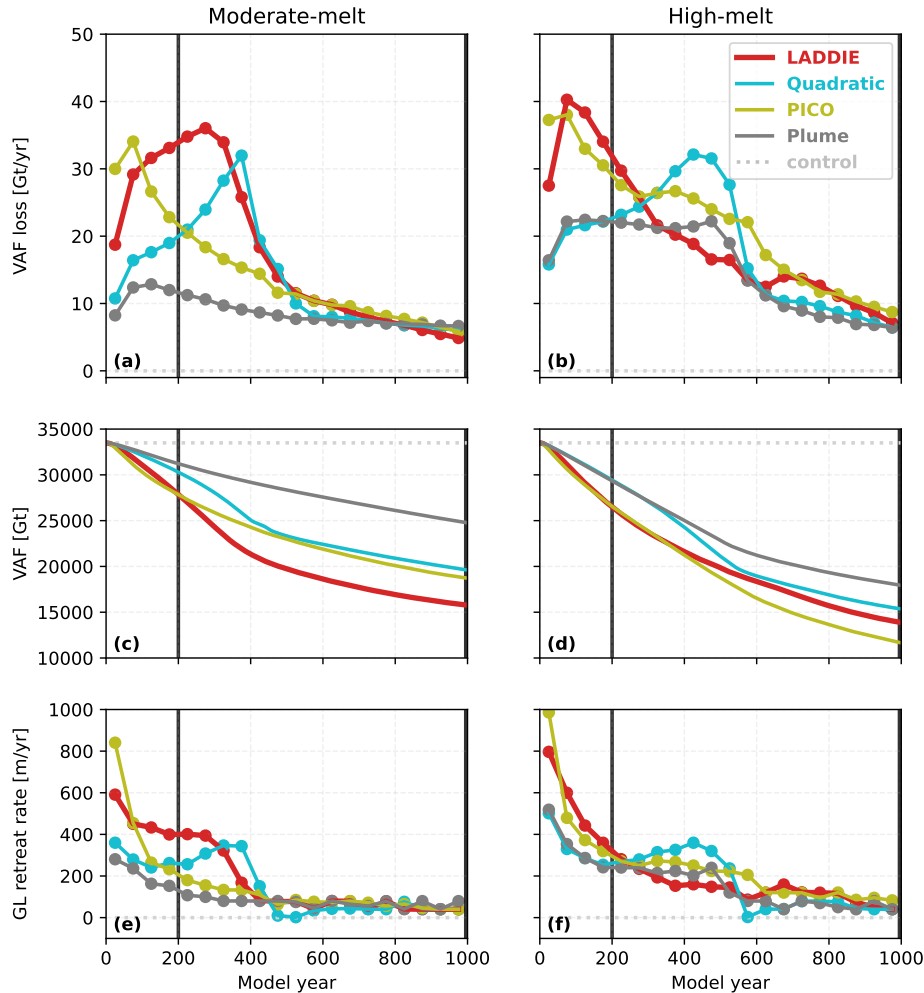

**Figure 7.** Transient response of scalar variables for the entire set of experiments. Volume above flotation (VAF) loss rates (a, b), VAF (c, d), and grounding line retreat rates for the central (Y = 0) grounding line (e, f) for the moderate-melt (left) and high-melt (right) experiments. VAF loss rates and grounding line retreat rates are calculated as the average rate of change within each consecutive 50-year interval. Thick vertical lines correspond to the time slices presented in Fig. 4 and Fig. 6. For the control experiment, only the VAF loss rates are shown.





This suggests that melt along the margins, which is strongly influenced by how the 2D horizontal meltwater flow is represented, may play a more significant role in determining grounding line retreat rates than melt near the grounding line.

On longer time scales, the distinction between the two groups disappears due to a later VAF loss peak of the Quadratic experiments, and the different stabilising mechanisms, discussed in Sect. 3.2.2, kicking in. These stabilising mechanisms cause VAF loss rates and grounding line retreat rates to converge to similar magnitudes across the different sub-shelf melt implementations after 600 years. On these time scales, larger-scale ice dynamics appear to be the primary driver of retreat and VAF loss, rather than the specific melt pattern, as the latter has little influence due to decreased thermal forcing.

For the resulting VAF at the end of the simulation, however, the choice of sub-shelf melt implementation matters. For the moderate-melt scenario, the melt pattern modelled by LADDIE leads to the largest response in terms of VAF loss over the course of the entire simulation (Fig. 7c). This loss is double that of the Plume experiment and about a quarter more than the losses observed in the Quadratic and PICO experiments. The feedback between steepening and enhanced deep melt rates, accompanied by substantial thinning and local collapse of the western margin, can sustain a long phase of high VAF loss rates

(Fig. 7a). For the Quadratic and PICO experiments, these periods of peak loss are substantially shorter, and for the Plume experiments, VAF loss rates never reach a similar magnitude. For the high-melt scenario, however, the PICO experiment gives the largest amount of VAF loss over the entire simulation (Fig. 7d). This is because the thermal forcing in the high-melt scenario maintains a wider collapse of the ice shelf margins compared to the moderate-melt scenario. As a result, dynamical detachment occurs over a wider area, leading to elevated VAF loss rates even at the end of the simulation (Appendix C, Fig. C2). In contrast,

the LADDIE experiment in the high-melt scenario still shows dynamical reattachment, as reduced thermal forcing weakens the meltwater circulation (Appendix C, Fig. C1). As a result, the final VAF loss in the PICO experiment is over 10% greater than that in the LADDIE experiment. The Plume experiment shows least VAF loss, however, the difference with LADDIE is reduced relative to the moderate-melt scenario.

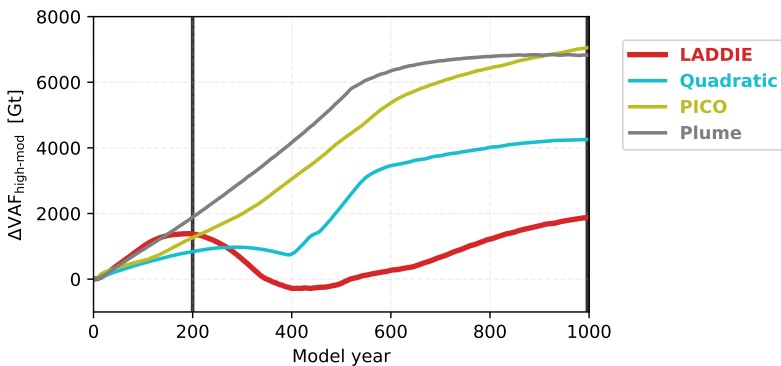

**Figure 8.** Sensitivity of volume above flotation (VAF) to the forcing scenario. Additional VAF loss in the high-melt scenario compared to the moderate-melt scenario: $\Delta\text{VAF}_{\text{high-mod}}$. Thick vertical lines correspond to the time slices presented in Fig. 4 and Fig. 6.



With melt along the margins influencing transient VAF loss and grounding line retreat, we argue it also plays a key role
in driving differences in melt sensitivity to the forcing scenario. The additional VAF loss in the high-melt scenario compared
to the moderate-melt scenario ($\Delta\text{VAF}_{\text{high-mod}}$) reveals that LADDIE shows a lower sensitivity to the imposed temperature
increase, compared to the parameterised experiments (Fig. 8). We attribute the low sensitivity in LADDIE to the difference
in the persistence of the western boundary channel between the moderate-melt and high-melt scenarios. In the moderate-
melt scenario, the channel persists longer, enabling a prolonged phase of dynamical detachment and reduced buttressing. In
contrast, in the high-melt scenario, amplified melt in the channel causes localised collapse, which disrupts the channel (negative
feedback), thereby shortening the period of reduced buttressing compared to the moderate-melt scenario (Appendix C).

For the parameterised experiments, melt along the margins determines the moment of repinning on the lateral margins
(Quadratic, Plume) and the extent of ice shelf collapse (PICO), which influence the VAF, and its sensitivity to ocean temper-
ature increase, through their effects on buttressing. For the Quadratic parameterisation, the sensitivity between the scenarios
increases sharply after 400 years. This coincides with the moment of pinning in the moderate-melt experiment, where increased
buttressing slows down the ice flow. In contrast, the high-melt experiment continues to experience sufficient melt, preventing
the ice from repinning just yet. For the PICO parameterisation, thermal forcing at depth at the central grounding line controls
melt rates along the margins. Higher temperatures at depth in the high-melt scenario cause widespread collapse, reducing but-
tressing and increasing VAF loss compared to the moderate-melt scenario. For the Plume parameterisation, early repinning in
the moderate-melt scenario allows the additional VAF loss in the high-melt scenario to accumulate until the high-melt simu-
lation itself also reaches the point of repinning. This again shows that the delay in possible repinning is linked to the stronger
VAF loss in a high-melt scenario.

In summary, melt along the margins influences the persistence of the western boundary current in LADDIE, the timing of
repinning in Quadratic and Plume, and the extent of collapse in PICO. As such, it is a key factor that governs the transient VAF
loss and grounding line retreat, and impacts the sensitivity to temperature increases. Since the magnitude and persistence of
margin melt is influenced by the way the 2D flow is represented, this representation of 2D flow is critical for both the transient
behaviour and sensitivity to increased forcing.

## 4   Discussion

Over the first centuries of our LADDIE experiments, the feedback between ice draft steepening and enhanced deep melt rates
strengthens the western boundary current, leading to a persistent thinning of the western margin. This margin thinning results
in a dynamical detachment between grounded and floating ice on the western side, which leads to reduced buttressing and
speed-up of the ice flow, resembling observations of the Crosson ice shelf (MacGregor et al., 2012; Goldberg et al., 2016;
Lilien et al., 2018). Furthermore, the western margin thinning aligns with previous modelling studies that identified amplified
melt rates on the Coriolis-favoured side of an ice shelf (Goldberg et al., 2012; Gladish et al., 2012). The period of sustained
weakening, coinciding with a period of strong volume loss, also supports findings from studies indicating that volume loss is
highly sensitive to shear-zone melt (Jordan et al., 2018; Feldmann et al., 2022). Hence, we believe that despite the schematic




character of our simulations, these observations of Crosson ice shelf can likely be explained by the 2D horizontal meltwater flow.

The phase of strong volume loss is followed by a phase of weaker volume loss as the meltwater circulation weakens and the ice shelf pins on the eastern margin, highlighting a distinct transient behaviour characterised by a rapid retreat followed by suppressed volume loss. The parameterisations either lack boundary melt (Quadratic, Plume), preventing a collapse of the western margin, or introduce melt through prescribed melting along the majority of the grounding line, hindering a recovery of the margins later in the simulation (PICO). These limitations in the parameterisations lead to different predictions of potential zones of ice shelf weakening, which might also apply when more realistic geometries are considered. Moreover, the parameterisations fail to replicate the transient behaviour observed in the LADDIE experiments, particularly in the timing and magnitude of peak volume loss. Altogether, this demonstrates that resolving the 2D horizontal meltwater layer dynamics leads to a distinct dynamical ice sheet response not captured by the parameterisations discussed.

Our LADDIE experiments replicate and expand upon key processes hypothesised and demonstrated in earlier studies. For example, Payne et al. (2007) used a model similar to LADDIE on a steady ice shelf draft of Pine Island Glacier and found high melt rates at the grounding line due to a strong ice draft steepness. They hypothesised that the positive feedback between steepening of the ice draft and higher melt rates would finally be compensated by the ice flow, which our simulations indeed confirm. Additionally, Sergienko (2013) showed that melt channel evolution occurs in the absence of variations in external forcing, driven solely by internal interactions between the ocean and the ice shelf. Similarly, in our experiments, we observe basal channel migration under constant external forcing.

The idealised framework of our experiments, both in geometry and forcing, necessitates caution when translating our results to realistic geometries. Our experimental geometry resembles the Pine Island ice shelf in terms of horizontal scale and position within grounded margins. For this ice shelf, melt along the western margin proved crucial in driving volume loss; however, this may not necessarily hold for ice shelves with wider grounding lines and multiple islands, such as the Getz ice shelf. In such cases, melt around these islands, which represents another form of margin melt, is likely to play a more significant role in governing buttressing (Selley et al., 2021). Consequently, differences between LADDIE and the parameterisations in calculating melt rates around such islands may be expected to have a greater impact on the differences in transient volume above flotation. The exact impact of these differences is beyond the scope of this study, as it depends on the specific characteristics of each ice shelf, requiring further investigation.

In terms of forcing, the abrupt transition from zero melt at initialisation to the initialised melt fields under the moderate-melt or high-melt scenario introduces a shock to the system. This abrupt change allows us to study the system's response time and resembles the transition from a cold to a warm cavity observed in several modelling studies (Naughten et al., 2021; Jin et al., 2024), or a sudden inflow of warm water due to unpinning (De Rydt et al., 2014) or calving (Bradley et al., 2022). The main caveat of the abrupt switch, compared to gradual temperature changes, is the relatively long time scales of ice dynamics. Ice flow cannot immediately compensate for ice loss driven by sub-shelf melt, allowing high melt rates to dominate the signal in the early years of the simulations. This leads to phenomena such as collapse of the ice shelf as a result of peak melt rates at locations with insufficient ice shelf thickness (Wearing et al., 2021). Some of these collapse locations partially recover during





the simulations but reopen later, highlighting their intrinsic sensitivity to melt. Altogether, the observations of the western margin collapse and speed up of the Crosson ice shelf indicate that this simulated process of collapse may also be induced by relatively moderate changes in ocean temperatures.

Various studies have emphasised the role of calving in influencing ice dynamics (Levermann et al., 2012; Benn and Åström, 2018), prompting us to assess its impact, which we found to be limited. We repeated our experiments, prescribing ice shelf calving when the ice thickness is below 100 meters, consistent with the MISOMIP1 protocol (Asay-Davis et al., 2016). While we recognise this as a simplified approach to representing true calving, accurately incorporating it into ice sheet models remains an active area of research (Wilner et al., 2023), so we use the threshold as a first-order approximation. Our sensitivity

experiments show that imposing a calving threshold has a limited impact on the transient behaviour for the LADDIE, Quadratic and Plume experiments (Appendix D). For the PICO experiments, we find a larger impact of calving on transient volume loss and grounding line retreat. For PICO, calving front retreat modifies the ice shelf-wide box structure and hence melt pattern. In contrast, for LADDIE, as well as for the Quadratic and Plume parameterisations, calving front retreat does not impact melting closer to the grounding line. In simulations with qualitatively different geometries, further investigation into the impact of

calving is needed, as it may have different effects in those cases.

Our results that ice sheet model simulations are sensitive to the representation of sub-shelf melt aligns with findings from previous studies. We observe a similar magnitude of variation as a result of this modelling choice, consistent with the findings of Berends et al. (2023). Furthermore, we highlight that a more physically advanced representation of sub-shelf melt does not always fall within the uncertainty range of the parameterisations, especially in our moderate-melt experiments. Favier et al.

(2019) found that the Plume parameterisation underestimates the 3D ocean model volume loss. Although their implementation of the Plume parameterisation differs from ours, we also find that our Plume experiments underestimate volume loss relative to our LADDIE experiments. Given that tuning targets can affect melt rates in scenarios beyond the tuned case (Burgard et al., 2022), we conducted additional parameterised experiments for the high-melt scenario, adjusting the deep melt rates to match those from the LA_H experiment (Appendix E). The results demonstrate that our main conclusions remain intact, suggesting

that the melt pattern plays a more critical role than the exact magnitude in determining the transient response. Finally, Lambert and Burgard (2024) showed that a consistent calibration between different sub-shelf melt parameterisations or models does not guarantee consistency in melt sensitivities. We add to this finding by showing that a consistent calibration also does not guarantee a consistent ice sheet response. Overall, our study further underscores that different sub-shelf melt implementations can lead to a divergent ice sheet response over various time scales, regardless of their initial calibration.

We want to emphasise that our coupled setup is not intended to fully replace coupled ice sheet-ocean models. Our setup captures changes in ice shelf base geometry leading to different melt patterns and vice versa. However, it does not take into account how changes in meltwater fluxes and geometry impact the general cavity circulation and, consequently, the ambient ocean forcing. De Rydt and Naughten (2024) showed that the evolution of the cavity geometry can substantially alter local ocean dynamics, thereby affecting sub-shelf melt rates. Although computationally demanding, these coupled simulations are

crucial for realistic long-term projections. In that context, our setup can be used to employ LADDIE's downscaling functionality by feeding it 3D ocean data to produce physically advanced 2D melt patterns on the ice sheet model grid. This can involve





offline-computed 3D ocean simulations or, in the future, online coupling between the ice sheet model and the 3D ocean model, with LADDIE serving as the interface. Besides these downscaling applications, our coupled setup also serves as a stand-alone tool for more process-based studies.

The reality is that many applications of ice sheet models still heavily rely on sub-shelf melt parameterisations (Seroussi et al., 2020, 2024; Hill et al., 2024), which, as our findings reveal, have key limitations compared to a more physically advanced sub-shelf melt representation, especially in how they represent 2D horizontal meltwater flow. These limitations result in different timing and location of ice shelf thinning, which are crucial for predicting which parts of the ice shelf are most vulnerable and when an ice shelf region may collapse. As a consequence, they also affect the transient behaviour of volume above flotation, which is essential for obtaining accurate sea level projections. To address these limitations, future research could focus on adapting parameterisations to better capture aspects of 2D horizontal meltwater flow, for instance by allowing meltwater plumes to follow the grounding line rather than following a straight path away from the grounding line (assumed in PICO, Plume). Furthermore, in upcoming model intercomparison projects such as ISMIP7, it would be valuable to include models that incorporate more physically advanced sub-shelf melt patterns. Given that our LADDIE experiment showed greater

volume loss over a millennial time scale under moderate melt forcing compared to the parameterised experiments, it is important to explore whether the feedback between steepening and enhanced melt along the grounding line and the western margin can also lead to volume loss beyond the parameterised range in more complex geometries with varying forcing.

## 5 Conclusions

The coupling of the sub-shelf melt model LADDIE and the ice sheet model IMAU-ICE offers a novel approach to incorporate
physically advanced sub-shelf melt patterns in fully dynamical ice sheet simulations. Our simulations conducted with this coupled setup reveal key differences to simulations using widely adopted sub-shelf melt parameterisations in terms of timing, persistence and the specific location of enhanced melt and eventual collapse of the ice shelf.

In an idealised geometry, the simulations with LADDIE reveal an important positive feedback mechanism that is not captured by the parameterisations when ocean conditions transition from a cold state with no melt to a warmer state. In both a moderate
and high warming scenario, the LADDIE experiments reveal an initial steepening of the ice draft near the grounding line, which strengthens the westward meltwater flow along the grounding line, further steepening the ice draft. This, in turn, enhances the western boundary current, leading to increased melt and progressive weakening of the western margin of the ice shelf. Parameterisations assume either no meltwater flow (Quadratic) or meltwater flow directed away from the grounding line (PICO, Plume), hence they cannot capture this feedback between ice draft steepening and enhanced melt rates along the central and
western grounding line.

This enhanced melt along the western margin governs the transient behaviour observed in the LADDIE experiments. Initially, thinning of the western margin facilitates a dynamic detachment of the grounded ice from the floating ice, reducing buttressing and accelerating velocities on centennial timescales. On millennial timescales, the meltwater flow weakens due to a reduced thermal forcing near the deep grounding line, suppressing the western boundary channel. This allows the western margin



ice shelf to thicken and dynamically reattach to the grounded ice, which ultimately reduces ice velocities and volume loss. Observations of western margin weakening indicate that this process may be important in the recent or future Antarctic response to changes in ocean forcing.

In contrast, the parameterisations cannot replicate this behaviour due to their limited representation of the 2D horizontal meltwater flow. The PICO experiments show high melt rates along the entire grounding line, persisting throughout the full

simulation. This forces an initial rapid retreat but prevents later thickening of the margins, thereby reducing buttressing compared to LADDIE. Meanwhile, the Quadratic and Plume experiments reveal low melt at the margins throughout the entire simulation, either due to the absence of meltwater flow (Quadratic) or underestimated flow convergence towards the margins (Plume) in these shallow areas. This limited margin melt constrains initial retreat, delaying peak volume loss (Quadratic) or reducing it substantially (Plume).

In summary, our findings suggest that resolving the 2D horizontal meltwater flow introduces new feedbacks and transient behaviour that sub-shelf melt parameterisations, which either neglect or approximate this flow, cannot replicate. The accurate representation of meltwater flow points to an explanation for the weakening of Coriolis-favoured sides of ice shelves and is therefore important to incorporate in more realistic geometries to improve estimates of the future evolution of the Antarctic ice sheet. This can be achieved either by refining existing parameterisations or by directly integrating more detailed melt patterns

generated by sub-shelf melt models that resolve the 2D horizontal meltwater flow.

*Code and data availability.* The code for the model versions used in this study, as well as the raw model output and post processed data is available via https://doi.org/10.5281/zenodo.14526103 (Jesse, 2024). The scripts to generate the plots are available on the Github page https://github.com/FrankaJes/meltwater_flow.

*Video supplement.* Animations showing the main variables for all perturbation experiments at a temporal frequency of 10 years are available

for download via the Github page https://github.com/FrankaJes/meltwater_flow.

## Appendix A: Details on sub-shelf melt parameterisations

This Appendix provides the implementation of the different sub-shelf melt parameterisations in IMAU-ICE.

### A1 Quadratic parameterisation

The implementation of the Quadratic parameterisation follows the approach of Favier et al. (2019). Melt rates are computed

based on a quadratic relation with the thermal forcing, using

$$m = \gamma_T \left( \frac{\rho_{\mathrm{w}} c_{\mathrm{po}}}{\rho_{\mathrm{i}} L_{\mathrm{i}}} \right) (T_{\mathrm{a}} - T_{\mathrm{f}})^2. \tag{A1}$$





The heat exchange coefficient, $\gamma_T$, is used as tuning parameter. $\rho_{\mathrm{w}}$ and $\rho_{\mathrm{i}}$ are the density of ocean water and ice respectively, $c_{\mathrm{po}}$ is the specific heat capacity of water, and $L_{\mathrm{i}}$ is the latent heat of fusion for ice. Values for these parameters are given in Table A1. $T_{\mathrm{a}}$ is the ambient ocean temperature at the depth of the ice shelf base for that particular grid cell. The local freezing temperature, $T_{\mathrm{f}}$, is defined as a function of salinity, $S$, and ice shelf base depth, $z_{\mathrm{b}}$, via

$$T_{\mathrm{f}} = \lambda_1 S + \lambda_2 - \lambda_3 z_{\mathrm{b}}. \tag{A2}$$

Here, $\lambda_1$, $\lambda_2$, and $\lambda_3$ are the liquidus slope, intercept and pressure coefficient, respectively (Table A1).

### A2   PICO parameterisation

The implementation of PICO follows Reese et al. (2018a). In our experiments, we use the setup with five ocean boxes. Melt rates are computed via

$$m_{\mathrm{k}} = -\frac{\gamma_T}{\nu \lambda}(a S_{\mathrm{a,k}} + b - c p_{\mathrm{k}} - T_{\mathrm{a,k}}), \tag{A3}$$

where the subscript k indicates the specific box, with corresponding melt, $m_{\mathrm{k}}$, and ambient temperature and salinity, $T_{\mathrm{a,k}}$ and $S_{\mathrm{a,k}}$, for that box. $\gamma_T$ is treated as a tuning parameter. Moreover, $\nu = \rho_{\mathrm{i}}/\rho_{\mathrm{w}}$, and $\lambda = L_{\mathrm{i}}/c_{\mathrm{po}}$, with these parameter values given in Table A1. The ambient ocean conditions $T_{\mathrm{a,k}}$ and $S_{\mathrm{a,k}}$ depend on the overturning flux $q$, e.g. how much water is transported from one box to the next. $p_{\mathrm{k}}$ is the overburden pressure, which introduces a depth dependency within each box. Parameters $a$, $b$, and $c$ take the same values as in Reese et al. (2018a). The overturning flux $q$ is defined by

$$q = C \rho_* (\beta(S_0 - S_1) - \alpha(T_0 - T_1)), \tag{A4}$$

with the overturning coefficient $C$ treated as a tuning parameter. $\rho_*$ is the reference density in the equation of state (EOS), given in Table A1. $T_0$ and $S_0$ are the salinity and temperature in box 0, and $T_1$ and $S_1$ in box 1.

### A3   Plume parameterisation

The implementation of the Plume parameterisations follows Lazeroms et al. (2018). Melt rates are computed via

$$m = M_0 \cdot g_1(\alpha) \cdot (T_{\mathrm{a}} - T_{\mathrm{f,gl}})^2 \cdot \sum_{k=0}^{11} p_{\mathrm{k}} \hat{X}^k. \tag{A5}$$

Here, $M_0$ is a constant parameter, listed in Table A1, the factor $g_1(\alpha)$ is a function which depends on the basal ice slope $\alpha$. $T_{\mathrm{a}}$ is the ambient ocean temperature parameter, $T_{\mathrm{f,gl}}$ is the freezing temperature at the plume origin grounding line. Coefficients $p_{\mathrm{k}}$ have the same value as reported in Lazeroms et al. (2018). And $\hat{X}$ is defined by

$$\hat{X} = \frac{z_{\mathrm{b}} - z_{\mathrm{gl}}}{l}, \tag{A6}$$

with $z_{\mathrm{b}}$ the local ice shelf draft depth, and $z_{\mathrm{gl}}$ the ice shelf draft depth at the grounding line. Length scale $l$ represents the distance from the grounding line at which melting transitions to refreezing, and is defined by

$$l = g_2(\alpha) \cdot \frac{T_{\mathrm{a}} - T_{\mathrm{f,gl}}}{\lambda_3}. \tag{A7}$$



It depends on the ambient ocean conditions, $T_a$, freezing temperature at the plume origin $T_{f,gl}$, and a slope-dependent factor $g_2(\alpha)$. This slope-dependent factor includes the heat exchange coefficient $\gamma_T$, which we treat as a tuning parameter. For the complete set of equations that describe the plume parameterisation and its implementation in IMAU-ICE, we refer to Lazeroms et al. (2018). To determine the plume origin, we use an approach similar to the 'average grounding-line origin' approach from Lazeroms et al. (2018). Instead of using the 16-direction search algorithm, we average over all grounding-line grid cells

following Berends et al. (2023).

| Symbol | Description | Value | Unit |
|---|---|---|---|
| $c_{po}$ | Specific heat capacity of ocean water | $3.97 \times 10^3$ | $\mathrm{J\,kg^{-1}\,K^{-1}}$ |
| $c_i$ | Specific heat capacity of ice | $2.01 \times 10^3$ | $\mathrm{J\,kg^{-1}\,K^{-1}}$ |
| $L_i$ | Latent heat of fusion for ice | $3.34 \times 10^5$ | $\mathrm{J\,kg^{-1}}$ |
| $\lambda_1$ | Liquidus slope | $-5.73 \times 10^{-2}$ | $\mathrm{°C\,psu^{-1}}$ |
| $\lambda_2$ | Liquidus intercept | $8.32 \times 10^{-2}$ | $\mathrm{°C}$ |
| $\lambda_3$ | Liquidus pressure coefficient | $7.61 \times 10^{-4}$ | $\mathrm{°C\,m^{-1}}$ |
| $\rho_i$ | Density of ice | 910 | $\mathrm{kg\,m^{-3}}$ |
| $\rho_w$ | Density of ocean water | 1028 | $\mathrm{kg\,m^{-3}}$ |
| $\rho_*$ | Reference density in EOS PICO | 1033 | $\mathrm{kg\,m^{-3}}$ |
| $M_0$ | Melt-rate parameter in Plume | 10 | $\mathrm{year^{-1}\,°C^{-2}}$ |
| $T_a$ | Ambient sea water temperature | - | $\mathrm{°C}$ |
| $S_a$ | Ambient sea water salinity | - | psu |
| $T_f$ | Pressure- and salinity-dependent freezing temperature | - | $\mathrm{°C}$ |

**Table A1.** Model parameters used in the sub-shelf melt parameterisations in IMAU-ICE. The same values are used for these variables in LADDIE.

# Appendix B: Model settings

Table B1 and B2 list the most important model settings for IMAU-ICE and LADDIE, respectively. Additionally, the parameter values shown in Table A1 are also applicable for LADDIE, meaning that both LADDIE and the parameterisations built into IMAU-ICE rely on the same values for these shared parameters. Complete model settings can be found in the configuration

files which can be accessed through the link in the data availability statement.




| Parameter | Definition | Value | Unit |
|---|---|---|---|
| $\alpha^2$ | Schoof law friction parameter | 0.5 | – |
| $\beta^2$ | Schoof law friction parameter | $1 \times 10^4$ | Pa m$^{-1/3}$ year$^{1/3}$ |
| $A$ | Flow factor | $1.44 \times 10^{-17}$ | Pa$^{-3}$ year$^{-1}$ |
| $T_i$ | Ice temperature | 270 | K |
| $\Delta x$ | Horizontal resolution | 2000 | m |
| $\Delta t_{min}$ | Minimal time step | 0.010 | year |
| $\Delta t_{max}$ | Maximal time step | 0.125 | year |

**Table B1.** Main parameter settings for the IMAU-ICE model setup used in this study. Details of the equations and the implementation of these parameters can be found in Berends et al. (2022) and Berends et al. (2023).

| Parameter | Definition | Value | Unit |
|---|---|---|---|
| $f$ | Coriolis parameter | $-1.37 \times 10^{-4}$ | m s$^{-1}$ |
| $T_i$ | Ice temperature | 270 | K |
| $U_{tide}$ | Tidal velocity | 0.1 | m s$^{-1}$ |
| $C_{d,mom}$ | Momentum drag coefficient | $2.5 \times 10^{-3}$ | – |
| $C_{d,top}$ | Top drag coefficient | $2.5 \times 10^{-3}$ | – |
| $\mu$ | Detrainment parameter | 2.5 | – |
| $A_h$ | Horizontal viscosity | 50 | m$^2$s$^{-1}$ |
| $K_h$ | Horizontal diffusivity | 50 | m$^2$s$^{-1}$ |
| $D_{min}$ | Minimum layer thickness | 1 | m |
| $\Delta x$ | Horizontal resolution | 2000 | m |
| $\Delta t$ | Time step | 92 | s |

**Table B2.** Main parameter settings for the LADDIE model setup used in this study. Details of the equations and the implementation of these parameters can be found in Lambert et al. (2023).

## Appendix C: High-melt scenario fields

As indicated in Sect. 3.1, the LADDIE experiments show a similar transient response in the high-melt scenario (LA_H) relative to the moderate-melt scenario (LA_M). One big difference is the duration of the phase of peak volume loss (Fig. 7a, b), which we find to be shorter in the high-melt scenario due to the negative feedback between collapse and western margin melt. This

negative feedback shows from the LA_H melt pattern and meltwater layer velocities in model year 200 (Fig. C1f, g). It shows the western boundary current cannot continue further along the western margin when it encounters one of the gaps in the ice shelf. Moreover, the eastward shift of the channelised meltwater flow occurs earlier in the high-melt scenario, allowing the channel to migrate fully to the east by the end of the simulation (Fig. C1k). Similar to the LA_M experiment, this channel is associated with low melt rates.



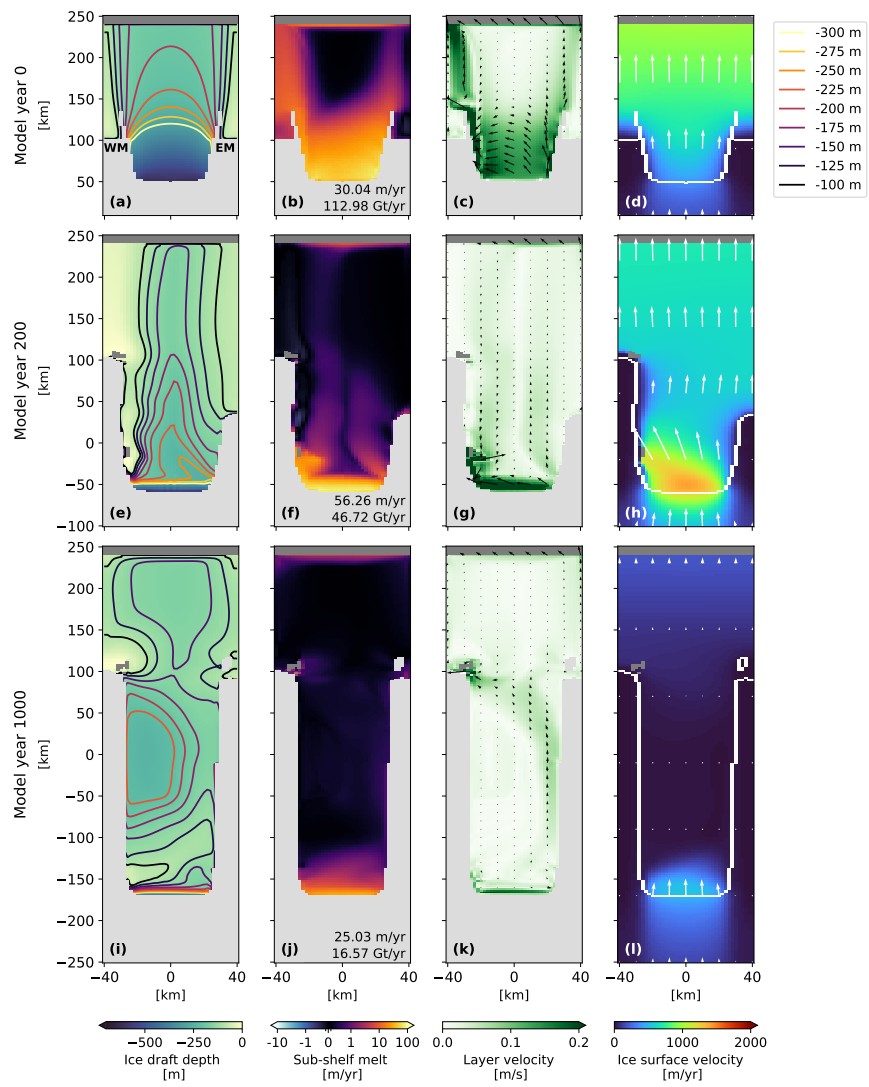

**Figure C1.** Same as Fig. 4, but now for high-melt scenario LADDIE experiment (LA_H).

The parameterised experiment under the high-melt forcing scenario show a similar feedback between melt and geometry
as seen under the moderate-melt forcing scenario (Fig. C2). Here, we highlight the main differences for each of the parame-
terisations. For the Quadratic parameterisation, in the high-melt experiment, melt rates remain to strong to facilitate pinning
on the margins at the end of the simulation (Fig. C2j,k), unlike in the moderate-melt experiment (Fig. 6j,k). The absence of
pinning results in weaker buttressing which leads to slightly higher final ice velocities at the end of the high-melt experiment
(Fig. C2l). For the PICO parameterisation, the high-melt scenario induces a wider collapse of the ice shelf (Fig. C2d), causing
strong dynamical detachment, leading to higher ice velocities than the moderate-melt scenario after 200 years (Fig. C2f). Over
longer timescales, enhanced melt rates lead to the near-complete loss of ice within the margins (Fig. C2m). Consequently,



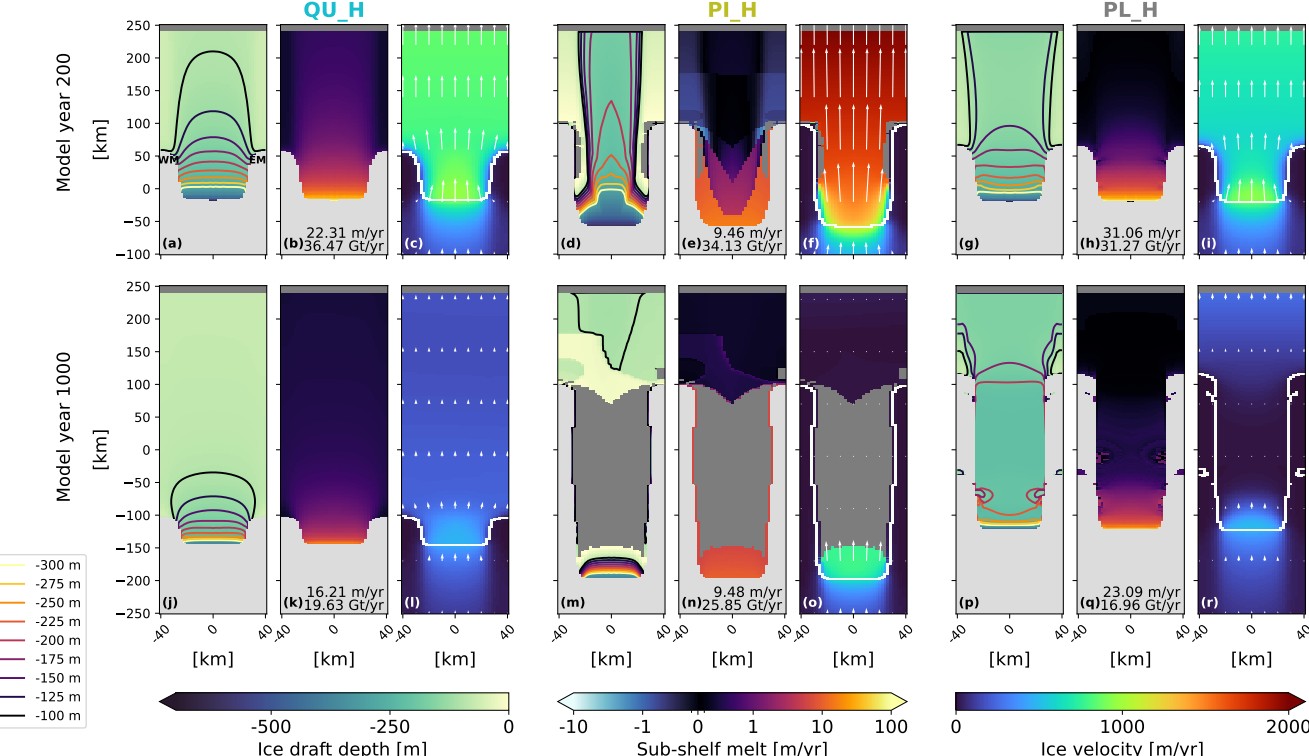

**Figure C2.** Same as Fig. 6, but now for the high-melt scenario Quadratic, PICO, and Plume experiments (QU_H, PI_H, PL_H).

ice velocities at the grounding line remain relatively high compared to the PI_M experiment (Fig. C2o). For the Plume pa-
rameterisation, the primary difference lies in the timing of margin pinning and readvance, which occur later in the high-melt

scenario compared to the moderate-melt scenario. This delay leads to a more retreated central grounding line by the end of the
simulation (Fig. C2r).

## Appendix D: Calving impact

To test the sensitivity to calving, all experiments were repeated with a calving threshold of 100 meters (i.e., floating ice is
removed when thinner than 100 meters). For LADDIE, and the Quadratic and Plume parameterisation, applying a calving

threshold has minimal impact on the transient behaviour (Fig. E1). The PICO experiments show a larger impact of calving,
primarily in the high-melt scenario, with volume loss rates revealing an earlier dip (years 200 to 600), but then a later peak
compared to the default experiments (years 600 to 800) (Fig. E1b). The stronger influence of calving in PICO is explained
by the calving front retreat modifying the ice shelf-wide box structure and melt pattern. For LADDIE, Quadratic and Plume,
calving front retreat does not affect melting closer to the grounding line.



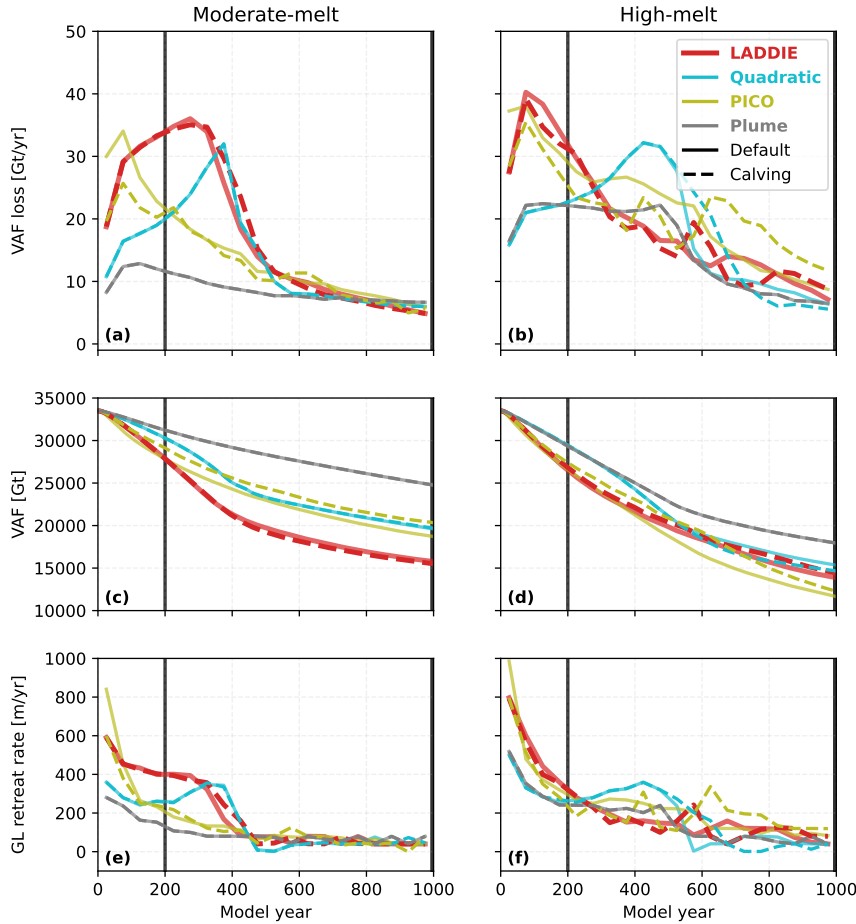

**Figure D1.** Same as Fig. 7, but now the dashed lines represent the simulations with calving applied using a threshold thickness of 100 meters.

## Appendix E: Retuning experiments


Table E1 shows the values of the tuning parameters for the default tuning scenario based on the moderate-melt scenario (same as Table 2), and for the retuned experiments, where we retuned the parameterisations to match deep melt rates with LADDIE in the high-melt scenario. For PICO, the overturning coefficient $C$ is kept constant over the two tuning scenarios.

For all parameterisations, retuning leads to lower integrated melt rates and deep melt rates at the start of the simulation.
For all parameterisations, retuning does not have a big effect on the transient behaviour over the first 200 years (Fig. E1). For the Quadratic and PICO parameterisations, retuning does not lead to major differences in transient VAF loss and grounding line retreat compared to the default tuning (Fig. E1b, g). This also leads to relatively similar VAF over time (Fig. E1d). For the Plume parameterisation, retuning to match deep average melt rates enables peak VAF loss rates to increase substantially compared to the default run, leading to stronger VAF loss and a substantial lower resulting VAF at the end of the simulation





| Sub-shelf melt option | default tuning $\gamma_T$ [ms$^{-1}$] | high-melt retuning $\gamma_T$ [ms$^{-1}$] | Overturning coefficient C [m$^3$ kg$^{-1}$] |
|---|---|---|---|
| LADDIE | $1.47 \times 10^{-4}$ | $1.47 \times 10^{-4}$ | |
| Quadratic | $14.16 \times 10^{-4}$ | $12.56 \times 10^{-4}$ | |
| PICO | $1.28 \times 10^{-4}$ | $0.93 \times 10^{-4}$ | $0.23 \times 10^{-6}$ |
| Plume | $16.87 \times 10^{-4}$ | $5.69 \times 10^{-4}$ | |

**Table E1.** Tuning parameter values for the different sub-shelf melt implementations. Default tuning refers to the tuning used for the main experiments presented in this study, with average deep melt rates of $16.75 \pm 1.00$ m/yr under the moderate-melt scenario. High-melt retuning refers to the tuning used for our retuning sensitivity tests, with average deep melt rates of $30.00 \pm 1.00$ m/yr under the high-melt scenario.

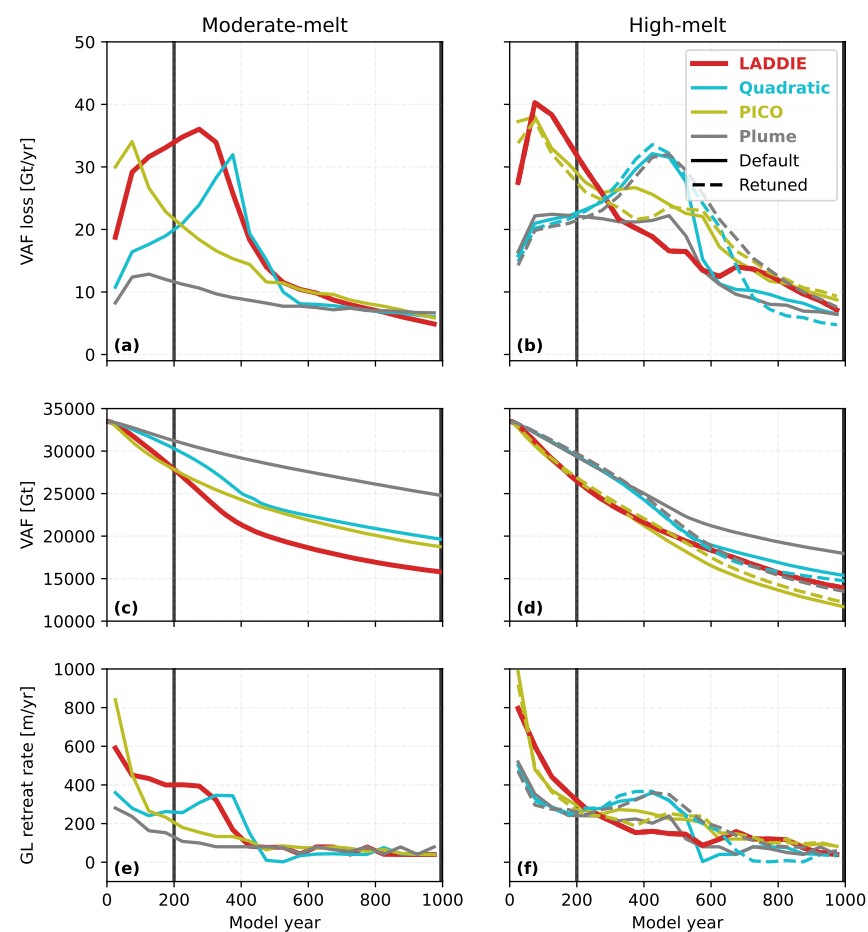

**Figure E1.** Same as Fig. 7, but now the dashed lines represent the simulations with melt rates retuned to match deep melt rate in high-melt scenario.



(Fig. E1d). The retuned run does not readvance on the lateral margins, whereas the default run does around model year 500. We suggest that this difference arises from the tuning affecting the length scale at which melt transitions to refreezing (Appendix A3). This length scale is shorter for the default run, allowing refreezing closer to the grounding line, consequently allowing for repinning.

## Appendix F:  Computational cost comparison

Computational costs are shown in Table F1. For the LADDIE experiments, IMUA-ICE runs on 31 cores, and LADDIE runs on 1 core. For the parameterised experiments, IMAU-ICE uses 32 cores. IMAU-ICE uses an iterative solver for the velocity fields, hence when geometries are more complex (e.g. pinning points), simulations can take longer. The simulations were run on an HPC with cores with variable efficiency.

| Experiment | Real time | CPUs |
|---|---|---|
| LA_M | 75 hours | 2400 CPUs |
| LA_H | 72 hours | 2304 CPUs |
| QU_M | 8 hours | 256 CPUs |
| QU_H | 24 hours | 768 CPUs |
| PI_M | 9 hours | 288 CPUs |
| PI_H | 11 hours | 352 CPUs |
| PL_M | 8 hours | 256 CPUs |
| PL_H | 9 hours | 288 CPUs |

**Table F1.** Overview on the computational cost of the perturbation experiments. Real time refers to the duration of a 1000-year simulation on one node with 32 cores.

*Author contributions.*  The conceptualisation was done by FJ, EL, and RvdW. FJ conducted the experiments and data analyses; produced the
figures; and provided the manuscript with input from EL and RvdW.

*Competing interests.*  The authors declare that they have no conflict of interest.

*Acknowledgements.*  FJ was funded by Utrecht University. EL was funded by the Netherlands Organization for Scientific Research (NWO) project HiRISE (grant no. OCENW.GROOT.2019.091). FJ used ChatGPT to enhance the readability of specific paragraphs. Following this, the content was reviewed and edited as necessary. Full responsibility is taken for the final publication.





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
