# Peer review of "Sub-shelf melt pattern and ice sheet mass loss governed by meltwater flow below ice shelves"

_EGUsphere, 2024_

## Referee Comment (RC1)

**Review of Jesse et al. "Sub-shelf melt pattern and ice sheet mass loss governed by meltwater flow below ice shelves"**

Reviewer: Xylar Asay-Davis

I wish my name to be relayed to the authors, as I feel I am always a better reviewer when I am not anonymous and I encourage others to consider reviewing non-anonymously whenever they feel able.

I also want to apologize to the authors for the lateness of my review. It was the result of a personal matter and in no way reflects a lack of enthusiasm for this work.

**General Comments:**

This manuscript describes a set of idealized experiments using a coupled ice sheet and 2D sub-ice-shelf boundary layer model (IMAU-ICE and LADDIE), comparing the LADDIE results with those using simpler parameterizations that are widely used in the ice-sheet modeling community. The experiments are based on the MISMIP+, ISOMIP+ and MISOMIP1 setup, using the same bedrock topography and initialization along with a qualitatively similar forcing approach. The main findings of this work are that the 2D representation of the sub-ice-shelf flow can capture significant physical processes that are missing from simpler parameterizations, and that these processes have a major impact on both the timing and the nature of ice-sheet mass loss and retreat. The authors make a compelling case that 2D plume models like LADDIE can provide substantial improvements in physical accuracy at high model resolution (2 km in this study) compared with simpler parameterizations while remaining substantially more affordable than 3D ocean models.

I found the results compelling and the paper to be well structured and well written. The figures and tables do an excellent job of supporting the paper. The choices of what material to put in the main text and what to present in appendices also seemed great to me. The numerous experiments are well designed and the results are compelling, and complementary to ongoing work in realistic configurations with these components. These findings are also complementary with the MISMIP+, ISOMIP+ and MISOMIP1 results, which are in various stages of publication.

I have a number of more specific comments as well as a few suggested grammatical and typographical corrections below. After some minor corrections, I think this paper will be ready for publication.

**Specific Comments:**

l. 17-19: "The parameterisations either inherently overestimate the persistence of margin thinning, leading to a sustained strong volume loss, or they underestimate margin thinning, delaying the onset of strong volume loss." In the main manuscript, I think you do an excellent job of providing enough context that it is clear why LADDIE provides the higher physical fidelity and thus can be postulated to be closer to the "true" solution. I would be careful here in the abstract, though. This sentence in particular makes it seem like the results with LADDIE are the truth that can be used to evaluate the deficiencies of the

parameterizations. I would strongly suggest you tone this down by adding something about "compared with LADDIE results" or something to that effect here.

sec. 2.1.1: I think it would be really good to include the 2-km horizontal resolution here somewhere. I know it's in Table B2 but that's pretty buried and this is a fairly fundamental feature of the modeling in both IMAU-ICE and LADDIE.

l. 91-93: "First, it was demonstrated that the choice of sliding law has little effect on the results for perturbation experiments in an idealised setup." I am concerned about this statement. It seems in direct contradiction to the findings of Cornford et al. (2020, https://doi.org/10.5194/tc-14-2283-2020), which found that the choice of basal friction led to the largest differences in model results in MISMIP+. There is an increasing consensus in the community (e.g. Joughin et al. 2019, https://doi.org/10.1029/2019GL082526) that Coulomb-limited friction laws like the Shoof parameterization that you use are both the most physically correct and the most consistent with observations. So I think it would be better to argue that you are using the "right" friction law, rather than that the friction law generally doesn't matter.

By the way, Cornford et al. (2020) would be an important paper to cite in your paper, since your setup is similar to MISMIP+ as you comment.

l. 121: I think "steady-state" needs a bit more explanation here even though you explain it later. In particular, LADDIE time-steps until it reaches steady state based on a given ice-sheet geometry and ambient ocean fields. It's important to make clear that the melt rates do evolve in time based on both ice-sheet geometry and (potentially, though not in this work) evolving ocean forcing.

l. 134-136: Could you say something about whether there is any coupling of heat fluxes?

l. 144-145 and caption of Fig. 1: "The coupling between the two models is asynchronous, meaning that draft geometry and sub-shelf melt rates are exchanged at a certain frequency: the coupling frequency." "The models are coupled asynchronously, meaning that they exchange geometry and melt rates at a fixed coupling frequency (in this case: 8 times per year) which is independent of the time step in the individual models."

There are different understandings in the community about what "asynchronous" means but this is not a definition I am familiar with. I have heard "asynchronous" used to describe running one model, then the other (which is what you do, so in that sense the term would be appropriate). But I have also heard the term used to describe running one component with an accelerated time compared with the other (e.g. running the ocean for 1 year but the ice sheet for 10 years) each time you couple. To me, "synchronous" coupling would be to run each model simultaneously for a coupling interval, then exchanging information. This is not what you do, but also involves coupling at a fixed coupling frequency. I do not think "asynchronous" coupling is typically used to refer to coupling less frequently than every model time step, which I think might be your understanding of the term.

l. 152-155: "The required runtime to reach a new quasi-steady-state depends on the flushing time…we run LADDIE for 4 days between each coupling step to ensure a

near-stable meltwater layer thickness and velocity." In my experience with ISOMIP+ Ocean0 (which is qualitatively similar to the beginning of your high-melt experiments), it takes several months for a 3D ocean model to reach quasi-steady state, suggesting a flushing time on the order of months. I believe you when you say that LADDIE reaches a new quasi-steady state in 4 days for this setup, but that suggests that it doesn't require anywhere near the cavity flushing time to do so. Instead, it suggests that the previous quasi-steady state was close enough to the new quasi-steady state to require only minimal adjustment, presumably over a time far less than the cavity flushing time. This is great because it saves you a lot of computation! But I think it means the cavity flushing time isn't the relevant time scale after all and the paragraph maybe needs to be revised accordingly.

l. 158-159: "To address this discrepancy, we use nearest neighbour averaging to extrapolate the resulting sub-shelf melt field to include the grounding line cells." Here is the part of the review where I tell you, perhaps unhelpfully, that this is not how I would have done things. I'm going to do that nonetheless because maybe we can have a discussion about it sometime. First, I'm a bit skeptical of the FCMP approach (even though I'm a co-author on the Leguy paper you cited for that). It seems like a low order choice from a mathematical perspective. But if that seems to be what works best, it is hard to argue with successful results. But regarding the extrapolation approach, wouldn't an alternative (maybe a preferable one) be to just have the LADDIE domain cover all cells with centers that are floating? And maybe even cover all cells that are even partially floating. You can always compute a melt rate in LADDIE but then use the fractional area that is floating as part of computing the total melt flux in the cell that you pass to IMAU-ICE. This could be done even if you stick with the FCMP approach.

l. 164: "and an upper limit of 0.125 years": Do you stick with 1/8 of a year even for your 50,000 year initialization simulation? If so, why (since you're not coupling)? If no, maybe state that since you are talking about initialization in this paragraph.

l. 158: "To obtain a stable central grounding line position at X = 50 km…" Please state somewhere in the text that you have defined your coordinate system differently (offset by 400 km in X and 40 km in Y) compared with MISMIP+, ISOMIP+ and MISOMIP1. Otherwise, the locations you refer to will be confusing to colleagues who are familiar with the original protocols and geometry for those experiments.

l.. 175-176; "We run the model for 40 days to obtain a quasi-steady-state melt pattern that aligns with the initial ice sheet geometry and forcing data." To me, this reinforces the comment above about the flushing time. To me, it seems like the flushing time must be on the order of 40 days for this setup with LADDIE, and that the subsequent 4-day LADDIE runs to quasi-steady state do not need to run for a full flushing time.

l. 190-191: "The salinity profiles are determined such that the density profiles of the moderate-melt and high-melt scenario are identical." I don't think you say anything in the paper about the equation of state you are using. Is it the same linear EOS as in Asay-Davis et al. (2016)? I think it may be necessary to mention that since you mention density profiles here.

l. 212-213 and 216: "...over the first period of the simulation…" "On longer time scales…": Especially because you only introduce time-series plots quite a bit later in Fig. 7, I think it would be important to give the reader a sense for what these times are – the first 200 to 400 years, and then the final 600 to 800 years would be my interpretation from Fig. 7.

Fig. 4: I found the gray pixels in panels e-h confusing at first.  I would explain them in the figure caption as areas of melt-through.

l. 244 and at least a dozen other places in the text: I believe when you use the term "collapse" here and in most other places in the text, you are referring to what I would call "melt-through".  To me (and I think to the broader ice-shelf community), the term "collapse" implies larger sections of the ice shelf breaking up by fracture, and is considered distinct from melt-through (though fracture would likely occur in reality before melt-through is possible).  I would ask you to change "collapse" to "melt-through" or a similar term except where you are referring to a larger break-up of the ice shelf by fracture.

l. 245-246: "When the western boundary current encounters one of these gaps in the ice shelf…" As written, I was confused by what was meant by "these gaps", since I did not understand "localized collapse" to cause gaps in the ice shelf but rather loss of larger sections.  In revising this paragraph to reword the "localized collapse" phrase above, please make sure that it is clear what "gaps" are being referred to here.

l. 246: "(assumption in LADDIE)": I think this could use a bit more explanation here.

l. 490-491: "This leads to phenomena such as collapse of the ice shelf as a result of peak melt rates at locations with insufficient ice shelf thickness (Wearing et al., 2021)." In addition to being another place where I would replace "collapse" with "melt-through", I would suggest rephrasing or explaining "insufficient".  I guess it seems like a tautology to me to say that melt-through happens at locations with insufficient ice-shelf thickness.

l. 503-504 and Appendix D: "...for LADDIE, as well as for the Quadratic and Plume parameterisations, calving front retreat does not impact melting closer to the grounding line." I was quite surprised by this.  My results for MISOMIP1 IceOcean2, which has a similar approach to calving, though it used a 3D ocean model, showed qualitatively different results with and without calving (see e.g. Fig. 12 in Asay-Davis et al. 2016).  I would have thought that 100-m thick ice would provide some fairly non-negligible buttressing in IMAU-ICE so that, even if melt rates didn't change appreciably, the flow of ice would be affected more substantially.  I just want to make sure the calving was applied in IMAU-ICE and not just used as a masking of melt in LADDIE.  Maybe IMAU-ICE just behaves very differently than BISICLES, the ice-sheet model we were using.

l. 513-514 and Appendix E: "...adjusting the deep melt rates to match those from the LA_H experiment (Appendix E)"  It wasn't clear to me either here or in Appendix E what exactly is different in this retuning.  My understanding was that the original tuning was done for the initial geometry such that the melt rate below -300 m was 30 m/yr.  Since this procedure was used for LADDIE, a retuning using the initial geometry wouldn't change anything.  So is the retuning for a different period of time?  Or what am I missing?  Could you describe the procedure and how it different from the initial tuning in more detail?

l. 520: "We want to emphasise that our coupled setup is not intended to fully replace coupled ice sheet-ocean models." I really appreciate you including this paragraph and this sentence in particular. I think it's an important caveat for the work.

l. 525-526: "In that context, our setup can be used to employ LADDIE's downscaling functionality by feeding it 3D ocean data to produce physically advanced 2D melt patterns on the ice sheet model grid." I really like this. This seems like a very powerful potential use for LADDIE and similar 2D boundary-layer models.

l. 539: Unless I missed it, ISMIP7 should be introduced as the "Ice Sheet Model Intercomparison Project for CMIP7". I think you can probably get away with not introducing CMIP7 but I will defer to the editors on that.

**Typographical, Grammatical and Formatting Suggestions:**

l. 15: I think "the parameterization's limitations" should probably be "the parameterizations' limitations" (i.e. both plural and possessive).

l. 113: Just a pet peeve of mine but maybe don't have back to back ") (" but instead use a semicolon.

l. 147: "...resulting into a coupling interval of 0.125 years." I think this phrase is obvious from the frequency of 8 times per year and can be removed.

l. 189: "tangent hyperbolic" should be "hyperbolic tangent"

l. 456-458: "Hence, we believe that despite the schematic character of our simulations, these observations of Crosson ice shelf can likely be explained by the 2D horizontal meltwater flow." I don't have any problem with this sentence scientifically – it's speculative but you make that clear. But I think there's a missing intermediate logical step here. May I suggest a rewording such as the following? "Although our simulations are schematic in character, we see qualitatively similar behavior. Hence, we believe that these observations of Crosson ice shelf can also likely be explained by the 2D horizontal meltwater flow." Still needs work but hopefully you get the point.

l. 460-461: "...characterised by a rapid retreat followed by suppressed volume loss." I think this phrase is redundant and can be removed.

---

## Author Response (AR2)

Dear editor,

Thank you for the opportunity to revise and resubmit our manuscript. We have carefully considered the comments from reviewer 1 and reviewer 2 and have made the necessary revisions to address their concerns. Below, you will find the original reply to the reviewers in green. Additionally, wherever our revision deviates from this reply, we provide an additional explanation in orange.

Furthermore, we have decided to move Appendix C to F to the supplementary material. With the extra appendix added for the cross-sections, we thought the Appendix was becoming too lengthy. We believe the supplementary material is well suited to support the discussion points but are not essential to understand the main story of the paper. We have retained the parameterisations' description and model parameters in Appendices A and B, as these are crucial for reproducibility and may be of immediate interest to readers.

In the final manuscript, we have also addressed the requested changes by the editor. Our responses to these can be found at the end of this document.

Best wishes,
Franka Jesse, Erwin Lambert, and Roderik van de Wal

Please find below our original reply to **Reviewer 1** in green.

**General Comments:**

This manuscript describes a set of idealized experiments using a coupled ice sheet and 2D sub-ice-shelf boundary layer model (IMAU-ICE and LADDIE), comparing the LADDIE results with those using simpler parameterizations that are widely used in the ice-sheet modeling community. The experiments are based on the MISMIP+, ISOMIP+ and MISOMIP1 setup, using the same bedrock topography and initialization along with a qualitatively similar forcing approach. The main findings of this work are that the 2D representation of the sub-ice-shelf flow can capture significant physical processes that are missing from simpler parameterizations, and that these processes have a major impact on both the timing and the nature of ice-sheet mass loss and retreat. The authors make a compelling case that 2D plume models like LADDIE can provide substantial improvements in physical accuracy at high model resolution (2 km in this study) compared with simpler parameterizations while remaining substantially more affordable than 3D ocean models.

I found the results compelling and the paper to be well structured and well written. The figures and tables do an excellent job of supporting the paper. The choices of what material to put in the main text and what to present in appendices also seemed great to me. The numerous experiments are well designed and the results are compelling, and complementary to ongoing work in realistic configurations with these components. These findings are also complementary with the MISMIP+, ISOMIP+ and MISOMIP1 results, which are in various stages of publication.

I have a number of more specific comments as well as a few suggested grammatical and typographical corrections below. After some minor corrections, I think this paper will be ready for publication.

Thank you for the positive feedback and for the constructive comments on how to further improve this manuscript. We agree with most of the comments and will implement them in the manuscript. Below, we provide a point-by-point response.

**Specific Comments:**

l. 17-19: "The parameterisations either inherently overestimate the persistence of margin thinning, leading to a sustained strong volume loss, or they underestimate margin thinning, delaying the onset of strong volume loss." In the main manuscript, I think you do an excellent job of providing enough context that it is clear why LADDIE provides the higher physical fidelity and thus can be postulated to be closer to the "true" solution. I would be careful here in the abstract, though. This sentence in particular makes it seem like the results with LADDIE are the truth that can be used to evaluate the deficiencies of the parameterizations. I would strongly suggest you tone this down by adding something about "compared with LADDIE results" or something to that effect here.

Agreed, we will rephrase this part in the abstract as following:
"This results in a different transient volume loss **between the parameterisations and LADDIE. Compared to LADDIE, the** parameterisations either inherently overestimate the persistence of margin thinning, leading to a sustained strong volume loss, or they underestimate margin thinning, delaying the onset of strong volume loss."

sec. 2.1.1: I think it would be really good to include the 2-km horizontal resolution here somewhere. I know it's in Table B2 but that's pretty buried and this is a fairly fundamental feature of the modeling in both IMAU-ICE and LADDIE.

In the submitted version of the manuscript, we mention the 2-km horizontal resolution in section 2.2.1. We agree, however, that it is good to mention this earlier in the model description so we will move the specification of horizontal resolution to 2.1.1. We will also add it to section 2.1.2, in which LADDIE is described. To avoid repetition, we will remove the comment on horizontal resolution from section 2.2.2.

l. 91-93: "First, it was demonstrated that the choice of sliding law has little effect on the results for perturbation experiments in an idealised setup." I am concerned about this statement. It seems in direct contradiction to the findings of Cornford et al. (2020, https://doi.org/10.5194/tc-14-2283-2020), which found that the choice of basal friction led to the largest differences in model results in MISMIP+. There is an increasing consensus in the community (e.g. Joughin et al. 2019, https://doi.org/10.1029/2019GL082526) that Coulomb-limited friction laws like the Shoof parameterization that you use are both the most physically correct and the most consistent with observations. So I think it would be better to argue that you are using the "right" friction law, rather than that the friction law generally doesn't matter.

By the way, Cornford et al. (2020) would be an important paper to cite in your paper, since your setup is similar to MISMIP+ as you comment.

We agree that the current phrasing may give the impression that the choice of sliding law has little influence on ice dynamics in general, which is not what we intended to communicate. To clarify, we will rephrase the paragraph as follows (or similar to this):

**"While the choice of sliding law has been shown to influence model results (Cornford et al., 2020), Berends et al. (2023), using the same ice sheet model and idealised domain as in our study, demonstrated that its impact is much smaller compared to that of the sub-shelf melt implementation. Given the focus on the latter, we conduct our experiments using a single sliding law: the Coulomb-limited modified power-law relation introduced by Schoof (2005). This choice is motivated by Joughin et al. (2019) which shows that Coulomb-limited laws best capture the dynamics at Pine Island Glacier."**

l. 121: I think "steady-state" needs a bit more explanation here even though you explain it later. In particular, LADDIE time-steps until it reaches steady state based on a given ice-sheet geometry and ambient ocean fields. It's important to make clear that the melt rates do evolve in time based on both ice-sheet geometry and (potentially, though not in this work) evolving ocean forcing.

Agreed, to clarify that this "steady-state" is connected to the combination of the ice shelf geometry and forcing at a given point in time, we will rephrase it as follows: "LADDIE is designed to simulate steady-state sub-shelf melt rates **for a given combination of ice shelf geometry and ambient ocean forcing.**"

l. 134-136: Could you say something about whether there is any coupling of heat fluxes?

The ice interior temperature influences ice shelf melt rates through the three-equation formulation. However, LADDIE does not modify the ice interior temperature, as there is no sensible heat flux into the ice at the ice-ocean interface. Consequently, the interior ice temperature remains constant throughout the simulation, following the MISOMIP1 protocol.

We will clarify this in Sect. 2.1.3 by adding something similar to:

**"Heat fluxes are coupled in one direction (output IMAU-ICE, input LADDIE). While the meltwater layer modelled by LADDIE is affected by the ice interior temperature through the three-equation formulation, the ice interior temperature itself is not affected by the meltwater layer, as there is no sensible heat flux into the ice at the ice-ocean interface."**

l. 144-145 and caption of Fig. 1: "The coupling between the two models is asynchronous, meaning that draft geometry and sub-shelf melt rates are exchanged at a certain frequency: the coupling frequency." "The models are coupled asynchronously, meaning that they exchange geometry and melt rates at a fixed coupling frequency (in this case: 8 times per year) which is independent of the time step in the individual models."

There are different understandings in the community about what "asynchronous" means but this is not a definition I am familiar with. I have heard "asynchronous" used to describe running one model, then the other (which is what you do, so in that sense the term would be appropriate). But I have also heard the term used to describe running

one component with an accelerated time compared with the other (e.g. running the ocean for 1 year but the ice sheet for 10 years) each time you couple. To me, "synchronous" coupling would be to run each model simultaneously for a coupling interval, then exchanging information. This is not what you do, but also involves coupling at a fixed coupling frequency. I do not think "asynchronous" coupling is typically used to refer to coupling less frequently than every model time step, which I think might be your understanding of the term.

Thanks for pointing this out. Our intention was to convey that the models run sequentially rather than simultaneously while also clarifying that LADDIE is not necessarily computed at every ice model time step.

We will remove 'asynchronous' from l.144, and the caption of Fig. 1. We believe that both the schematic in Fig. 1 and the description in the second paragraph of Sect. 2.1.3 clearly convey that the models run sequentially (as you also inferred). Therefore, we will avoid using 'asynchronous coupling' anywhere in the text to prevent potential confusion.

l. 152-155: "The required runtime to reach a new quasi-steady-state depends on the flushing time...we run LADDIE for 4 days between each coupling step to ensure a near-stable meltwater layer thickness and velocity." In my experience with ISOMIP+ Ocean0 (which is qualitatively similar to the beginning of your high-melt experiments), it takes several months for a 3D ocean model to reach quasi-steady state, suggesting a flushing time on the order of months. I believe you when you say that LADDIE reaches a new quasi-steady state in 4 days for this setup, but that suggests that it doesn't require anywhere near the cavity flushing time to do so. Instead, it suggests that the previous quasi-steady state was close enough to the new quasi-steady state to require only minimal adjustment, presumably over a time far less than the cavity flushing time. This is great because it saves you a lot of computation! But I think it means the cavity flushing time isn't the relevant time scale after all and the paragraph maybe needs to be revised accordingly.

We agree that the cavity flushing time is not the relevant time scale to refer to in this context. As you suggest, a more appropriate reference is the flushing time of the meltwater layer, which represents a much smaller volume and typically involves higher velocities than the full ocean cavity. This results in a substantially shorter flushing time.

Using the total volume of the meltwater layer divided by total entrainment, we estimate flushing times of approximately 23 days in the moderate-melt scenario and 18 days in the high-melt scenario — consistent with the 40-day spinup phase required from rest.

As you point out, subsequent 4-day LADDIE runs do not need to span a full flushing time. Rather, the adjustment to a new quasi-steady state is much faster, due to only minor changes in geometry between coupling steps. To clarify, we will remove the flushing time from l. 152–155. We will reword this as follows:

**"The time required for LADDIE to reach a new quasi-steady state after each coupling step depends on the magnitude of changes in geometry and external forcing. Since we keep the oceanic forcing constant throughout our experiments, the adjustment time is driven by changes in geometry. In our setup, we found that running LADDIE for 4 days between coupling steps of 0.125 years is sufficient for the meltwater layer thickness and velocity to reach near-stable conditions."**

We will add the meltwater layer flushing time as the relevant time scale to spin up LADDIE from rest (see our response to your comment below).

l. 158-159: "To address this discrepancy, we use nearest neighbour averaging to extrapolate the resulting sub-shelf melt field to include the grounding line cells." Here is the part of the review where I tell you, perhaps unhelpfully, that this is not how I would have done things. I'm going to do that nonetheless because maybe we can have a discussion about it sometime. First, I'm a bit skeptical of the FCMP approach (even though I'm a co-author on the Leguy paper you cited for that). It seems like a low order choice from a mathematical perspective. But if that seems to be what works best, it is hard to argue with successful results. But regarding the extrapolation approach, wouldn't an alternative (maybe a preferable one) be to just have the LADDIE domain cover all cells with centers that are floating? And maybe even cover all cells that are even partially floating. You can always compute a melt rate in LADDIE but then use the fractional area that is floating as part of computing the total melt flux in the cell that you pass to IMAU-ICE. This could be done even if you stick with the FCMP approach.

We see that we incorrectly stated this in the manuscript. LADDIE considers grid cells to be within the ice shelf mask when the ice shelf draft at the cell center exceeds the bedrock height at the cell center, thus satisfying the FCMP condition. We will correct this in the revised manuscript.

The nearest neighbour averaging is still implemented to address a practical issue that arises in our coupled setup: between coupling time steps, grounding line retreat can cause new cells to become afloat. While the sub-shelf melt parameterisations are immediately updated to reflect this change, LADDIE only updates its domain at the next coupling step. Without extrapolating melt rates to these newly floating cells, they would temporarily lack any melt input, potentially affecting the ice dynamics. However, in our experiments, the time step of the ice sheet model was equal to the coupling time step during the largest part of the simulation. As a result, the extrapolated melt was rarely applied in practice. We will clarify this in the revised manuscript.

l. 164: "and an upper limit of 0.125 years": Do you stick with 1/8 of a year even for your 50,000 year initialization simulation? If so, why (since you're not coupling)? If no, maybe state that since you are talking about initialization in this paragraph.

We used an upper limit of 0.125 years for the 50,000-year initialisation as well. While this was not strictly necessary, it remained in the configuration file and was used in the simulation. Since this detail is not specific to the initialisation, we have decided to move it—along with the horizontal resolution—to sect. 2.1.1 for better clarity.

l. 158: "To obtain a stable central grounding line position at X = 50 km…" Please state somewhere in the text that you have defined your coordinate system differently (offset by 400 km in X and 40 km in Y) compared with MISMIP+, ISOMIP+ and MISOMIP1. Otherwise, the locations you refer to will be confusing to colleagues who are familiar with the original protocols and geometry for those experiments.

Thanks for pointing this out. We will add the following to the beginning of this paragraph (sect. 2.2.1):

"The ice sheet model is initialised following the MISMIP+ protocol (Asay-Davis et al., 2016). **Compared to the geometry presented in the protocol paper, the coordinate system is offset by 400 km in the X-direction and 40 km in the Y-direction (Fig. 2).**"

We will also clarify it in the caption of Fig. 2.

l. 175-176; "We run the model for 40 days to obtain a quasi-steady-state melt pattern that aligns with the initial ice sheet geometry and forcing data." To me, this reinforces the comment above about the flushing time. To me, it seems like the flushing time must be on the order of 40 days for this setup with LADDIE, and that the subsequent 4-day LADDIE runs to quasi-steady state do not need to run for a full flushing time.

Yes, that is correct — the meltwater layer flushing time is on the order of 40 days (see our response to the comment above). This time scale is relevant during the initial spin-up from rest, while subsequent runs require much shorter adjustments due to only minor changes in geometry. To clarify, we will revise lines 175–176:

**"We run the model for 40 days to obtain a quasi-steady-state melt pattern that aligns with the initial ice sheet geometry and forcing data. This time scale of 40 days equals approximately twice the flushing time of the meltwater layer, which depends on both the size of the ice shelf and the oceanic forcing. This meltwater layer flushing time is considerably shorter than the flushing time of the entire cavity due to the smaller volume of the meltwater layer and the higher velocities within it (Holland et al., 2017)."**

l. 190-191: "The salinity profiles are determined such that the density profiles of the moderate-melt and high-melt scenario are identical." I don't think you say anything in the paper about the equation of state you are using. Is it the same linear EOS as in Asay-Davis et al. (2016)? I think it may be necessary to mention that since you mention density profiles here.

Yes, we used the same linear EOS coefficients as in Asay-Davis et al. (2016), which we will list in Table A1 and reference in the text. These coefficients were applied consistently for computing the forcing in our experiments.

It is important to emphasise that the original PICO paper (Reese et al., 2018) used different EOS coefficients. For the PICO parameterisation, we retained those

coefficients to maintain consistency with Reese et al. (2018). We will also add them to Table A1.

While double-checking, we discovered that the salinity profiles shown in Fig. 3 were based on the PICO linear EOS coefficients from Reese et al. (2018). However, the forcing applied in the experiments was correctly computed using the EOS coefficients from Asay-Davis et al. (2016). We will update Fig. 3 to show the salinity profiles corresponding to the actual forcing used.

l. 212-213 and 216: "…over the first period of the simulation…" "On longer time scales…": Especially because you only introduce time-series plots quite a bit later in Fig. 7, I think it would be important to give the reader a sense for what these times are – the first 200 to 400 years, and then the final 600 to 800 years would be my interpretation from Fig. 7.

We will follow your suggestion and rephrase:

"over the first **300 years** of the simulation", "On longer time scales **(600 to 800 years)**"

Fig. 4: I found the gray pixels in panels e-h confusing at first. I would explain them in the figure caption as areas of melt-through.

To clarify this, we will add the following to the figure caption:

"In all panels, the **ice-free** ocean is masked by dark grey. **Dark grey pixels enclosed by ice shelf pixels indicate areas of melt-through.**"

l. 244 and at least a dozen other places in the text: I believe when you use the term "collapse" here and in most other places in the text, you are referring to what I would call "melt-through". To me (and I think to the broader ice-shelf community), the term "collapse" implies larger sections of the ice shelf breaking up by fracture, and is considered distinct from melt-through (though fracture would likely occur in reality before melt-through is possible). I would ask you to change "collapse" to "melt-through" or a similar term except where you are referring to a larger break-up of the ice shelf by fracture.

Yes, we agree, we will follow your suggestion to replace all mentions of "collapse" by "**melt-through**", except for two occasions where we refer to possible break-up of the ice shelf not solely caused by melt (l. 36, l. 534).

l. 245-246: "When the western boundary current encounters one of these gaps in the ice shelf…" As written, I was confused by what was meant by "these gaps", since I did not understand "localized collapse" to cause gaps in the ice shelf but rather loss of larger sections. In revising this paragraph to reword the "localized collapse" phrase above, please make sure that it is clear what "gaps" are being referred to here.

We will change "gaps" to "**areas of melt-through**" to clarify this.

l. 246: "(assumption in LADDIE)": I think this could use a bit more explanation here.

We agree that this explanation should be expanded. Areas of melt-through are treated the same as ice-free cells at the calving front. At these boundaries, the pressure gradient force drives a strong outflow of the meltwater across the calving front. Any momentum, heat, or salt advected beyond this boundary is lost to what is treated as an infinite open ocean.

To clarify this, we will add a paragraph to section 2.1.2 (LADDIE description) providing a more detailed explanation of how these conditions are handled in the model.

l. 490-491: "This leads to phenomena such as collapse of the ice shelf as a result of peak melt rates at locations with insufficient ice shelf thickness (Wearing et al., 2021)." In addition to being another place where I would replace "collapse" with "melt-through", I would suggest rephrasing or explaining "insufficient". I guess it seems like a tautology to me to say that melt-through happens at locations with insufficient ice-shelf thickness.

We agree, so we will rephrase it as follows:

"**This leads to phenomena such as melt-through of the ice shelf, which occurs when the applied melt rate exceeds the ice shelf thickness divided by the ice sheet model time step (Wearing et al., 2021).**"

l. 503-504 and Appendix D: "...for LADDIE, as well as for the Quadratic and Plume parameterisations, calving front retreat does not impact melting closer to the grounding line." I was quite surprised by this. My results for MISOMIP1 IceOcean2, which has a similar approach to calving, though it used a 3D ocean model, showed qualitatively different results with and without calving (see e.g. Fig. 12 in Asay-Davis et al. 2016). I would have thought that 100-m thick ice would provide some fairly non-negligible buttressing in IMAU-ICE so that, even if melt rates didn't change appreciably, the flow of ice would be affected more substantially. I just want to make sure the calving was applied in IMAU-ICE and not just used as a masking of melt in LADDIE. Maybe IMAU-ICE just behaves very differently than BISICLES, the ice-sheet model we were using.

The calving is indeed applied in IMAU-ICE, not just used as a masking in LADDIE. We will clarify this in the manuscript.

We can distinguish between two main potential impacts of calving in these simulations:
(1) Changes in buttressing, and
(2) Changes in melt rates near the deep grounding line.

Regarding (1), we observe minimal impact of calving on buttressing in the coupled LADDIE–IMAU-ICE simulations. This is because the 100-meter thickness threshold causes the loss of ice primarily along the western margin, which already contributes

very little to buttressing in the default (no calving) configuration. Removing it entirely does not substantially affect the upstream ice.

For (2), calving can influence melt rates depending on how the sub-shelf melt implementation handles the geometry. For PICO, calving significantly alters the box configuration, consequently impacting melt rates near the grounding line. In contrast, LADDIE, Quadratic and Plume show minimal/no change in deep melt rates under calving. In a 3D ocean model, the ocean circulation could respond more strongly to calving front retreat – affecting melt rates near the grounding line.

We will clarify this distinction between the two effects in the revised manuscript to improve understanding of our results.

l. 513-514 and Appendix E: "...adjusting the deep melt rates to match those from the LA_H experiment (Appendix E)" It wasn't clear to me either here or in Appendix E what exactly is different in this retuning. My understanding was that the original tuning was done for the initial geometry such that the melt rate below -300 m was 30 m/yr. Since this procedure was used for LADDIE, a retuning using the initial geometry wouldn't change anything. So is the retuning for a different period of time? Or what am I missing? Could you describe the procedure and how it different from the initial tuning in more detail?

The original LADDIE tuning was performed for the initial geometry, with the averaged melt rates below -300 m set to 30 m/yr in the high-melt scenario, resulting in 16.75 m/yr in the moderate-melt scenario. The parameterisations were then adjusted to match the deep melt rates in the moderate-melt scenario, as this is the scenario we discuss most extensively.

This is explained in section 2.2.3, l. 203: "We then take the resultant averaged deep melt rates in the moderate-melt scenario to tune the parameterisations."

However, the scenario used for tuning the parameterisations may seem somewhat arbitrary. To address this, we also conducted retuning experiments, where the parameterisations were tuned using the 30 m/yr average melt rate from the high-melt scenario. These experiments are presented in Appendix E. We understand this could be confusing, so we will clarify it further in the Appendix by specifying in Table E that the default tuning is based on the moderate-melt scenario.

l. 520: "We want to emphasise that our coupled setup is not intended to fully replace coupled ice sheet-ocean models." I really appreciate you including this paragraph and this sentence in particular. I think it's an important caveat for the work.

Yes, we agree that this is important to mention.

l. 525-526: "In that context, our setup can be used to employ LADDIE's downscaling functionality by feeding it 3D ocean data to produce physically advanced 2D melt

patterns on the ice sheet model grid." I really like this. This seems like a very powerful potential use for LADDIE and similar 2D boundary-layer models.

We fully agree and we are looking forward to working on this application in the future!

l. 539: Unless I missed it, ISMIP7 should be introduced as the "Ice Sheet Model Intercomparison Project for CMIP7". I think you can probably get away with not introducing CMIP7 but I will defer to the editors on that.

Agreed, we will include the full name for ISMIP7.

**Typographical, Grammatical and Formatting Suggestions:**

l. 15: I think "the parameterization's limitations" should probably be "the parameterizations' limitations" (i.e. both plural and possessive).

Agreed, we will follow your suggestion.

l. 113: Just a pet peeve of mine but maybe don't have back to back ") (" but instead use a semicolon.

We will change it, and we will move the name to the first occurrence of the LADDIE abbreviation in the introduction.

l. 147: "…resulting into a coupling interval of 0.125 years." I think this phrase is obvious from the frequency of 8 times per year and can be removed.

Agreed, we will remove it.

l. 189: "tangent hyperbolic" should be "hyperbolic tangent"

Thanks, we will change this.

l. 456-458: "Hence, we believe that despite the schematic character of our simulations, these observations of Crosson ice shelf can likely be explained by the 2D horizontal meltwater flow." I don't have any problem with this sentence scientifically – it's speculative but you make that clear. But I think there's a missing intermediate logical step here. May I suggest a rewording such as the following? "Although our simulations are schematic in character, we see qualitatively similar behavior. Hence, we believe that these observations of Crosson ice shelf can also likely be explained by the 2D horizontal meltwater flow." Still needs work but hopefully you get the point.

Thanks for this suggestion, we agree that this intermediate step improves the readability. We will rephrase this sentence as follows in the updated manuscript:

**"Although our simulations are schematic in terms of geometry and forcing, we see qualitatively similar behaviour. Hence, we believe that these observations of Crosson ice shelf can likely be explained by the 2D horizontal meltwater flow."**

l. 460-461: "...characterised by a rapid retreat followed by suppressed volume loss." I think this phrase is redundant and can be removed.

Agreed, we will remove it.

Please find below our original reply to **Reviewer 2** in green. In addition, wherever our revision deviates from this reply, we provide an additional explanation in orange.

**General Comments:**

The paper compares a coupled model setup (involving an ice sheet model coupled to a 2D single layer ocean model) to more commonly used parameterisations for calculating ocean-induced melt under a floating ice shelf. Both qualitative and quantitative differences are shown to be significant. The significance of the western boundary current in particular is highlighted, and its impact on buttressing through enhanced thinning of shear margins. This is an important, though not surprising, result, and advances the science of ice sheet – ocean interactions. The overall layout and the clarity of the text is excellent. I recommend the manuscript to be published with minor modifications.

Thank you for the positive feedback and for the constructive comments on how to further improve this manuscript. We agree with most of the comments and will implement them in the manuscript. Below, we provide a point-by-point response.

What I most missed in this study was a comparison with a 3D ocean model. Although coupling IMAU-ICE to a 3D ocean circulation model might be beyond the scope of this study, circulation patterns from ISOMIP+ and MISOMIP1 ocean simulations should be obtainable through those projects. It is clear that LADDIE had a stronger physical justification than the other parameterisations presented here, but does it's 2D pattern look similar to 3D ocean models for the same domain? I'm not going to insist on this for the current paper, but I do hope the authors will find an opportunity to compare LADDIE against some of the ISOMIP+/MISOMIP1 models. It is worth at least pointing out that the LADDIE description paper includes a comparison against mitgcm for a real world ice shelf.

Indeed, coupling IMAU-ICE to a 3D ocean model lies beyond the scope of this study. While we emphasise that the coupling to LADDIE is not meant to replace coupled ice-ocean models (l.520), we agree that such a comparison would be valuable and is also in line with the comparison presented in Lambert et al. (2023). Therefore, we will include a paragraph in the discussion section describing how LADDIE compares to the results presented in the references at the bottom of this document (ISOMIP+/MISOMIP1 studies).

Given that a significant component of the ice sheet retreat shown in this study is attributable to marine ice sheet instability, I think this needs at least a paragraph (or at least a sentence) at some point. Presumable the imposed melt triggers MISI, and the ensuing retreat and VAF loss is due to a combination of melt-induced thinning and MISI.

We will add a paragraph on MISI in section 3.2.3 to highlight its role in the initial phase of retreat. Specifically, we agree that the imposed melt likely triggers MISI, which contributes to the early grounding line retreat before it crosses the bedrock depression.

In most simulations, this crossing occurs within 200 years and generally coincides with the peak rates of grounding line retreat.

Are there real world ice shelve cavities where a neutrally buoyant layer separates from the underside of the shelf at some depth? Presumably LADDIE would be valid for such cavities?

In LADDIE, the separation of the neutrally buoyant layer (for instance at Filchner Ronne) is represented through gradual detrainment, allowing a substantial portion of the meltwater to detach from the upper layer while maintaining a minimum upper layer thickness across the domain. We will clarify this treatment of detrainment in section 2.1.2 (LADDIE description).

The "collapse" of a part of the ice shelf on the west side needs more attention, both in terms of describing exactly how this is handled in LADDIE and whether this implementation is a physically realistic. More specifically, what are the implications for momentum and heat loss and is this what would happen in the real world?

We will elaborate on the implications for momentum and heat loss in the methods section as you suggest in one of the line-by-line comments below. Find a more detailed response in how we plan to do that in our response to that comment.

There's a lack of explanations of what the abbreviations actually stand for. E.g. LADDIE, PICO etc. Do these stand for something? I'm sure LADDIE does, not sure about PICO? And IMAU-ICE? Please mention the full names when you first introduce the abbreviations. [Edit: I see the LADDIE full name appears on line 113; please move it earlier]

We will move the LADDIE full name up to the introduction and we will add the full name of PICO. We will also add the full name for ISMIP7 at the end of the discussion.

The authors choose a power law sliding equation and quote a reference indicating that choice of sliding equation doesn't have a large impact. I haven't read this paper, and I am happy to accept that it is correct in the context of the referenced paper, but it is very clear that, in general, choice of sliding equation can have a huge impact on simulated marine ice sheet behaviour.

We agree that the current phrasing may give the impression that the choice of sliding law has little influence on ice dynamics in general, which is not what we intended to communicate.

We will rephrase this paragraph to something similar to:

 "**While the choice of sliding law has been shown to influence model results (Cornford et al., 2020), Berends et al. (2023), using the same ice sheet model and idealised domain as in our study, demonstrated that its impact is much smaller compared to that of the sub-shelf melt implementation. Given the focus on the**

**latter, we conduct our experiments using a single sliding law: the Coulomb-limited modified power-law relation introduced by Schoof (2005). This choice is motivated by Joughin et al. (2019) which shows that Coulomb-limited laws best capture the dynamics at Pine Island Glacier."**

Related to this, Figure 4i seems to have a very steep gradient in the ice shelf close to the grounding line. Grounded ice geometry is not shown, but I presume that a very steep thickness gradient exists across the grounding line? A sliding equation that represents the dependency of basal resistance on effective pressure, which must decrease toward zero in the vicinity of the grounding line, is unlikely to exhibit this kind of feature. I would like to see at least one plot showing ice thickness, or ice upper surface height, also for grounded ice upstream of the grounding line. This could also be in supporting material or an appendix, and doesn't need to be shown for all simulations. Just something to indicate the shape of the ice approaching the grounding line. I'd also like to see some comment on the potential of different sliding equations to impact on the results.

Indeed, the surface gradient close to the grounding line is very steep in Fig. 4i, also shown by the geometry cross-sections for the same time slices as in Fig. 4 (see below). We will include this figure in the supplementary material.

We acknowledge that there may be an interaction between the melt pattern and the choice of sliding law, and we will add a paragraph to the discussion to reflect on this potential influence, also reflecting on the figure below.

[Figure]

**Line by line comments:**

13. "first period" is not meaningful to the reader at this point. Are we talking about a few decades here? Please make this a bit clearer to the reader who has not yet read the experiment design.

Agreed, we will replace "first period" by "300 years".

15. Suggest "parameterisation" -> "simpler parametrisation" or "more common parameterisation" because to my mind the 2D LADDIE could be viewed as a parameterisation (albeit a more sophisticated one) so it wasn't obvious to me that LADDIE is not being discussed here.

Although the application of LADDIE is, in this study, comparable to the application of the parameterisations, the fundamental nature of LADDIE is different. It is a regional 2D numerical model which integrates a set of conservation equations in time. We believe the classification of parameterisation does not do justice to this complexity, and hence prefer to retain the distinction between parameterisations and LADDIE.

85-86. I'm missing a reference to a description of the SIA/SSA model as implemented here. The Berends paper describes the DIVA implementation in IMAU-ICE and states that there is also a hybrid SSA/SIA model in IMAU-ICE, but doesn't describe it. Please include a reference to a paper that describes the SSA/SIA hybrid as it is implemented here. Perhaps a Beuler PISM paper?

Thanks for pointing this out. The SIA/SSA model is implemented in IMAU-ICE following the same method as Beuler & Brown 2009 indeed, so we will add the reference there.

155. How have you determined that 4 days is enough to spin up the 2D ocean from rest? This seems like a very short time to me!

A full spin up from rest is only performed during the initial spin up phase of LADDIE (step 2 in Fig. 1b), where LADDIE requires roughly 40 days to equilibrate. This time scale equals approximately twice the flushing time of the meltwater layer. This meltwater layer flushing time is considerably shorter than the flushing time of the entire cavity due to the smaller volume of the meltwater layer and the higher velocities within it. We will add this to the LADDIE spin up paragraph (l.175-176).

In the coupling phase (step 4 in Fig. 1b), LADDIE is not spun up from rest; instead, it is restarted from the equilibrium state obtained in the previous coupling step. Since the forcing remains constant, and the changes in geometry between coupling intervals are small due to the short coupling time step, the model only requires 4 days to adjust and to reach near-stable conditions. We will clarify this in the manuscript.

Fig. 2. Would be nice to also see the spun up LADDIE state.

The spun up state of LADDIE is identical to the state at t = 0, and therefore already shown in both Fig. 4 (melt rates and velocity) and Fig. 5 (only melt rates). To avoid repetition and to focus on the initial ice sheet geometry, we prefer to leave the spun up state of LADDIE out of Fig. 2.

236-243. I didn't understand this first time through. I think you're saying that in general the central flow is being held back by lateral transmission of stress across the shear margins, so if this transmission was removed the central part would flow faster and the narrow strip along each side would flow slower. So the "collapse" (thinning to zero) of ice in the western shear margin allows the central flow to increase, and the strip along the side to slow down. Is that right?

Indeed, the central flow is affected by lateral transmission of stress across the margins. For the western side, this transmission is mostly removed, hence the central flow can speed up compared to the eastern side.

Additionally, because of the slow-down of the grounded strip, the east-west oriented grounding line at the very edge of the domain (which we referred to as WM grounding line) cannot retreat further. This contrasts with the EM grounding line, where the decoupling of the velocity field is not apparent.

To avoid confusion, we will explicitly clarify the role of lateral stress transmission across the shear margins in relation to the dynamical detachment of the velocity field.

I think I was initially confused by the labelling, direction and interpretation of "margin". Given that "margin" could plausibly refer to the margin of the domain itself, and that "WM" and "EM" are horizontal in the figure, I was confused between the short section of east-west grounding line at the very edges of the domain and the approximately north-south sections of grounding line close to the shear margins within the domain. I was thinking of the former when I think you were referring to the latter. Can you align the text with the grounding line that you intend to refer to? And can you clarify exactly what you mean by "western margin" and "WM grounding line"? I guess "margin" means either "shear margin" or "cavity margin" and not "domain margin". You could also edit the text in paragraph 223-228. This is also relevant for correctly understanding your description at line 340.

Regarding the terminology, we understand the confusion around the use of the word "margin" and how it may be interpreted either as the domain boundary or as a shear margin. In our manuscript, by "margin" we specifically mean "shear margin", not the outer boundary of the model domain.

The "WM" and "EM" grounding lines refer to the east-west oriented grounding lines located at the edges of the domain. These are the features we intended to emphasise, so your initial interpretation was correct. We recognise, however, that the dynamical detachment also occurs along the north-south oriented section of the western shear margin.

We agree that the current definitions are unclear and may contribute to misinterpretation. In the revised manuscript, we will address this by:
- Renaming ('WM' and 'EM' → '$GL_W$' and '$GL_E$' ) and clearly introducing the east-west oriented grounding lines in both the text and the figure
- Refering to the **'western and eastern shear margins'** when we discuss the north-south oriented shear margins.

Fig 4. It is very hard to see the Layer velocity arrows. The speed comes across fine, but, except for the top plot, it is hard to interpret the direction of flow. The flow pattern does not look complex, so perhaps it would be ok to have fewer arrows and scale them up a bit?

We will remove some arrows and scale them up to better visualise the flow direction.

We now use arrows of the same length for both the plume velocity and the ice surface velocity (i.e. they only indicate direction and not magnitude). For the plume velocity, we only show arrows in grid cells where the velocity is 0.02 m/s or higher.

Is there a reason to stop at 300 with the ice draft contours? Come to think of it, is there a reason to start at 100? Also, I find it a bit confusing that you have both a full colour plot and contours, each using a different colorscale, for what appears to be the same quantity (I think these are both for the ice draft if I understood right?)

Yes, both the full color plot and contours represent ice shelf draft. We use the explicit contours to better visualise the asymmetry. We chose to stop at 300 m because it marks the boundary of our tuning area. The current range provides a clear overview without overwhelming the plot with too many lines.

We agree that the different colorscales can be confusing, so we will consider a non-linear colorscheme which highlights this range better for the revised manuscript. We have replaced the original contours with solid black lines for the -100, -200, and -300 contour lines, and we now use dashed lines for 50 m intervals in between.

I think you can simply say "draft" instead of "draft depth".

We will replace 'draft depth' by 'draft' throughout the text and in the figures.

There is dark grey shown on the left side of the panels in the second row. I'm not quite sure what this is – perhaps am ice – ocean mask discrepancy? Can you clarify this in the caption please? [Edit: having read further I think this might be when the ice shelf vanishes completely; i.e. zero draft? The meaning of "the ocean is masked by dark grey" is not completely obvious because LADDIE is of course representing some aspect of the ocean]

Indeed, these dark grey patches on the left side indicate locations of melt-through (where the ice shelf completely vanishes). We will, also considering the last part of the above comment, rephrase to the following:

"In all panels, the **ice-free** ocean is masked by dark grey. **Dark grey pixels enclosed by ice shelf pixels indicate areas of melt-through.**"

The ice flow looks pretty stagnant in 4l through much of the shelf. Maybe a log scale would show the ice flow better?

We prefer to use a linear scale for the ice velocity plots, as it more clearly shows the differences between the different sub-shelf melt options and time slices. The ice flow in Fig. 4l is indeed quite stagnant, and this is shown to emphasise the low velocities at the end of the simulations. In the revised manuscript (l.260), we will explicitly state the contrast between these stagnant velocities at the end of the simulation and the previous time slices shown.

244. Can you be clearer what you mean by "collapse"? It looks like a very thin ice shelf, but I don't know if it actually goes to zero?

These are areas where the applied melt rate exceeds the ice shelf thickness divided by the ice sheet model time step, effectively removing all ice and resulting in ice-free conditions (i.e., zero ice thickness). We now realise that 'collapse' is a confusing term here. Therefore, we will change 'collapse' to 'melt-through' throughout the text, except for two occasions where we refer to possible break-up of the ice shelf not solely caused by melt (l. 36, l. 534).

246. I think "the buoyant plume exits the cavity" and "assumption in LADDIE" is not quite sufficient explanation here. I suggest you add a paragraph in the model description describing boundary conditions in LADDIE and how vanishing ice shelf is handled. I guess you impose zero normal velocity in LADDIE at the lateral (east and west) boundaries? Then what exactly do you do when ice shelf vanishes? I haven't read the LADDIE paper and I don't know what it's state variable(s) are (I guess T, S, and velocity?). Can you write down the boundary condition equations used at the transition to a vanishing shelf? Or is it not handled as a boundary as such, in which case, what exactly happens?

The areas of melt-through are treated identically to ice-free cells at the calving front. At these transitions from ice-covered to ice-free cells, we prescribe zero gradients in the state variables temperature (T), salinity (S), layer thickness (D), and horizontal velocity components (U, V).

In the momentum equations, the horizontal pressure gradient force assumes zero draft for open ocean cells. This drives a strong outflow of the meltwater across the calving front. Any momentum, heat, or salt advected beyond this boundary is lost to what is treated as an infinite open ocean. Hence, momentum, heat, and salt cannot be transferred across an ice-free cell — this is what we meant when referring to the plume "exiting the cavity."

We agree that this treatment deserves a clearer explanation in the manuscript. As you suggest, we will add a dedicated paragraph in section 2.1.2 (LADDIE description). We will also refer to this when discussing the results.

247. Well presumably sea ice would form in the real world, limiting heat loss? The Southern shear margin in the PIG ice shelf might be pretty similar to this situation?

Indeed, in reality, sea ice formation could limit heat loss to the atmosphere and potentially influence the meltwater layer dynamics. We will add a reflection on this in the discussion section, linking it to LADDIE's treatment of ice-free areas (see our response above). This treatment may be more limiting for isolated melt-through holes than for open ocean conditions at the calving front, and we will clarify this distinction accordingly.

Fig. 7. "For the control experiment, only the VAF loss rates are shown" What does this mean? It looks to me like you show VAF and GL retreat for the control also…

Thanks for pointing this out, this was left from an older version of the figure. We will remove this part from the figure caption.

456-457. What does "schematic character" mean? Do you mean the idealised MISOMIP domain?

Here, 'schematic character' indeed points to the idealised domain (or geometry) as well as to the idealised ocean forcing. We will clarify this in the manuscript and replace:

"Hence, we believe that despite the schematic character of our simulations these observations of Crosson ice shelf can likely be explained by the 2D horizontal meltwater flow."

by:

"**Although our simulations are schematic in terms of geometry and forcing, we see qualitatively similar behaviour. Hence, we believe that these observations of Crosson ice shelf can likely be explained by the 2D horizontal meltwater flow.**"

545. "advanced" doesn't have a specific meaning here. How about "physically motivated" instead?

We agree, we will follow your suggestion and change it to 'motivated' instead.

553. Suggest "Parameterisations" -> "Commonly used parameterisations".

We will follow your suggestion to emphasise that Quadratic, PICO and Plume are among the commonly used sub-shelf melt parameterisations.

**REFERENCES**

- Zhao, C., Gladstone, R., Galton-Fenzi, B., Gwyther, D., & Hattermann, T. (2022). Evaluation of an emergent feature of sub-shelf melt oscillations from an idealised coupled ice-sheet/ocean model using FISOC (v1. 1)-ROMSIceShelf (v1. 0)-Elmer/Ice (v9. 0). *Geoscientific Model Development Discussions*, *2022*, 1-27.

- Vaňková, I., Asay-Davis, X., Branecky Begeman, C., Comeau, D., Hager, A., Hoffman, M., … & Wolfe, J. (2025). Subglacial discharge effects on basal melting of a rotating, idealized ice shelf. *The Cryosphere*, *19*(1), 507-523.

- Scott, W. I., Kramer, S. C., Holland, P. R., Nicholls, K. W., Siegert, M. J., & Piggott, M. D. (2023). Towards a fully unstructured ocean model for ice shelf cavity

environments: Model development and verification using the Firedrake finite element framework. *Ocean Modelling, 182*, 102178

- Gwyther, D. E., Kusahara, K., Asay-Davis, X. S., Dinniman, M. S., & Galton-Fenzi, B. K. (2020). Vertical processes and resolution impact ice shelf basal melting: A multi-model study. *Ocean Modelling, 147*, 101569.

- Favier, L., Jourdain, N. C., Jenkins, A., Merino, N., Durand, G., Gagliardini, O., ... & Mathiot, P. (2019). Assessment of sub-shelf melting parameterisations using the ocean–ice-sheet coupled model NEMO (v3. 6)–Elmer/Ice (v8. 3). *Geoscientific Model Development, 12*(6), 2255-2283.

Please find below our reply to the editor's corrections **in green**.

L440-445: Melt-driven retreat down a retrograde slope should not be confused with MISI. To show MISI, as done by e.g. Rosier et al. 2021 and Reed et al. 2024, a dedicated set of experiments is needed to demonstrate that no stable locations of the grounding line exist. In fact, Gudmundsson et al. 2012 showed that for this domain and geometry, the grounding line can be stable on the retrograde slope.

We appreciate this observation and have revised the paragraph to clarify the distinction between melt-driven retreat and MISI, as well as pointing out that the grounding line can be stable on the retrogade slope – thereby nuancing the role of MISI in our simulations (it could be playing a role but not necessarily). It now reads as follows:

"In all perturbation experiments, the onset of sub-shelf melting **potentially triggers** marine ice sheet instability (MISI) over the first period of the simulations. Initially, the grounding line is positioned on a retrograde slope, which changes to a prograde slope beyond X = 0 km (Fig. 2). Until the grounding line advances across this shift in bed slope, VAF loss and grounding line retreat **are** driven by a combination of melt-induced thinning and MISI. In most experiments, this crossing occurs within the first 200 years, also being the period which holds the peak rates of VAF loss and grounding line retreat. **We note, however, that for a similar geometry, stable grounding line positions can also exist (Gudmundsson et al., 2012). This suggests that MISI may not occur across the entire retrograde bed slope, and in some areas, retreat may be driven only by sub-shelf melting.** Once the grounding line crosses the bedrock depression at X = 0 km, MISI no longer occurs, and subsequent retreat is governed solely by continued thinning due to melting."

L533-540: You currently allow the ice thickness to become zero in originally ice-covered regions, at least in LADDIE (also in IMAU-ICE?). You could test the sensitivity to this choice and preserve the heat and momentum all the way to the ice front by imposing a minimum ice thickness in both models. I'm not suggesting this as an additional simulation for this paper, but it could be something to look at in the future.

We allow the ice thickness to become zero in originally ice-covered regions in both IMAU-ICE and LADDIE. For future experiments, applying a minimum ice thickness in originally ice-covered areas may be helpful in clarifying the role of the negative feedback between melt-through events and the disruption of the boundary channel. Thank you for this suggestion.

L565-575: Although the reviewer has asked for cross sections of ice draft and a discussion about the sensitivity of the basal slope to the choice of basal sliding law, I'm not sure how much you can say about that without doing simulations with different sliding laws. The statement "In contrast to power-law sliding laws such as that of Weertman (1957), the Coulomb-limited Schoof sliding law used in our study yields stronger surface gradients across the grounding line, which may influence the melt–geometry feedback." either needs a reference or should be shown explicitly from

model simulations with different sliding laws. I'm not sure why one would expect "a flatter draft … under power-law sliding".

Upon rereading both the reviewer's comment and your comment above, we realise that we misinterpreted the concern. The Schoof sliding law we use depends on the effective pressure, which approaches zero near the grounding line. From previous simulations in the same geometry (Berends et al., 2023), we find that the Weertman sliding law, which does not depend on effective pressure, produces slightly steeper surface gradients across the grounding line. The draft gradient just downstream of the grounding line is slightly steeper, albeit marginally. In light of this, we have rephrased the paragraph as follows:

"**Using sliding laws other than the Coulomb-limited Schoof sliding law used in our study could yield different surface gradients across the grounding line, which may influence the melt–geometry feedback. For example, sliding laws that do not depend on effective pressure, such as the Weertman power-law relation (Weertman et al., 1957) may lead to a steeper ice shelf draft near the grounding line. This occurs because they cause a less pronounced reduction in sliding in that region compared to effective pressure-dependent laws like the Schoof sliding law used here. While Schoof sliding already produces a steep draft (see Fig. S3), sensitivity tests suggest that the draft under Weertman sliding could be slightly steeper, albeit marginally (Berends et al., 2023).**
**In LADDIE, a steeper draft could increase meltwater layer velocities at the central grounding line and along the western shear margin. However, the melt–geometry feedback described in Section 3.1 is not expected to change fundamentally. Increased meltwater layer velocities may cause faster thinning of the western shear margin and, consequently, earlier melt-through events that disrupt the boundary channel. The latter could shorten the rapid retreat phase via the negative feedback between melt-through events and margin melting discussed in Section 3.1. Since earlier sensitivity tests showed only minor differences in draft steepness, switching to a different sliding law is unlikely to substantially affect the melt–geometry feedback for this particular geometry.**
For the parameterisations, Berends et al. (2023) showed that for this geometry, the choice of sliding has little effect on final volume compared to the choice of sub-shelf melt parameterisation. However, the choice of sliding law, in combination with more complex geometries, warrants further investigation into the melt-geometry feedback derived from LADDIE and the various parameterisations."

L601 and following: how do you define and diagnose buttressing here? Do you look at the instanteneous change in the velocities when ice is removed?

We did not conduct a quantitative analysis of buttressing ourselves. In this paragraph, the emphasis is on the qualitative aspects of the buttressing effect rather than its quantitative measurement. We have clarified this in the text as follows:

"Second, calving can reduce ice shelf buttressing **if it occurs in key areas, such as the shear margins, which play a critical role in buttressing (Fürst et al., 2016)**. In PICO,

although calving **along the shear margins** may reduce buttressing, its effect is outweighed by the accompanying changes in melt distribution. In the Quadratic and Plume experiments, calving is minimal **and confined to the ice shelf front, resulting in a negligible impact** on buttressing. In the LADDIE experiments, calving primarily affects ice along the western margin. However, in our default setup, this ice already contributes little to buttressing due to dynamical detachment. As a result, calving, implemented here via a simple thickness threshold, has a limited impact on transient behaviour in LADDIE for this idealised geometry."